# A Theoretical View on Sparsely Activated Networks

**Cenk Baykal**
Google Research

**Nishanth Dikkala**
Google Research

**Rina Panigrahy**
Google Research

**Cyrus Rashtchian**
Google Research

**Xin Wang**
Google Research

## Abstract

Deep and wide neural networks successfully fit very complex functions today, but dense models are starting to be prohibitively expensive for inference. To mitigate this, one promising direction is networks that activate a sparse subgraph of the network. The subgraph is chosen by a data-dependent routing function, enforcing a fixed mapping of inputs to subnetworks (e.g., the Mixture of Experts (MoE) paradigm in Switch Transformers). However, prior work is largely empirical, and while existing routing functions work well in practice, they do not lead to theoretical guarantees on approximation ability. We aim to provide a theoretical explanation for the power of sparse networks. As our first contribution, we present a formal model of data-dependent sparse networks that captures salient aspects of popular architectures. We then introduce a routing function based on locality sensitive hashing (LSH) that enables us to reason about how well sparse networks approximate target functions. After representing LSH-based sparse networks with our model, we prove that sparse networks can match the approximation power of dense networks on Lipschitz functions. Applying LSH on the input vectors means that the experts interpolate the target function in different subregions of the input space. To support our theory, we define various datasets based on Lipschitz target functions, and we show that sparse networks give a favorable trade-off between number of active units and approximation quality.

## 1 Introduction

Overparameterized neural networks continue to yield performance gains as their sizes increase. This trend has been most prominent with large Transformer based language models [6, 12, 32]. However, using large, dense networks makes training and inference very expensive, and computing a forward pass may require trillions of floating point operations (FLOPs). It is an active area of research to improve the scalability and efficiency without decreasing the expressiveness or quality of the models.

One way to achieve this goal is to only activate part of the network at a time. For example, the Mixture of Experts (MoE) paradigm [22, 37] uses a two-step approach. First, each input is mapped to a certain subnetwork, known as an expert. Then, upon receiving this input, only this particular subnetwork performs inference, leading to a smaller number of operations compared to the total number of parameters across all experts. Some variations of this idea map parts of an input (called tokens) individually to different experts. Another well-studied generalization is to map an input to a group of subnetworks and aggregate their output in a weighted manner. Switch Transformers [14] successfully use a refined version of the MoE idea, where the input may be the embedding of a token or part of a hidden layer's output. Researchers have investigated many ways to perform the mapping, such as Scaling Transformers [21] or using pseudo-random hash functions [34]. In all cases, sparsity occurs in the network because after the input is mapped to the subnetwork, only this subset of neurons

needs to be activated. Moreover, the computation of the mapping function, sometimes referred to as a 'routing' function, takes significantly less time than the computation across all experts.

The success of these approaches is surprising. Restricting to a subnetwork could instead *reduce* the expressive power and the quality of the model. However, the guiding wisdom seems to be that it suffices to maximize the number of parameters while ensuring computational efficiency. Not all parameters of a network are required for the model to make its prediction for any given example. Our goal is to provide a theoretical explanation for the power of sparse models on concrete examples.

**Our Results.** Our first contribution is a formal model of networks that have one or more sparsely activated layers. These sparse layers are data dependent: the subset of network activations and their weight coefficients depend on the input. We formally show that our model captures popular architectures (e.g., Switch and Scaling Transformers) by simulating the sparse layers in these models.

We next prove that the above class of sparsely activated models can learn a large class of functions more efficiently than dense models. As our main proof technique, we introduce a new routing function, based on locality sensitive hashing (LSH), that allows us to theoretically analyze sparsely activated layers in a network. LSH maps points in a metric space (e.g., $\mathbb{R}^d$) to 'buckets' such that nearby points map to same bucket. The total number of buckets used by a LSH function is referred to as the size of the hash table. Prior work has already identified many routing functions that work well in practice. We do not aim to compete with these highly-optimized methods, but rather we aim to use LSH-based routing as a way to reason about the approximation ability of sparse networks.

In Theorem 4.1, we show that LSH-based sparse models can approximate real-valued Lipschitz functions in $\mathbb{R}^d$. Although real data often lives in high dimensions, the underlying manifold where the inputs are supported is often assumed to be low-dimensional (e.g. the manifold of natural images vs $\mathbb{R}^{224 \times 224}$ for a $224 \times 224$ image). We model this by assuming that our inputs lie in a $k$-dimensional manifold within $\mathbb{R}^d$. To get $\epsilon$ approximation error, we need an LSH table of size approximately $O((\sqrt{dk}/\epsilon)^k)$ but a forward pass only requires time $O(dk \log(1/\epsilon))$ as only one of the $O((\sqrt{dk}/\epsilon)^k)$ non-empty buckets are accessed for any given example.

In Theorem 4.3, we complement our upper bounds by proving a lower bound of $\Omega((2\sqrt{d}/\epsilon)^k)$ on the size needed for both dense and sparse models (when points live in $k$ dimensions). This also transforms into a lower bound on the number of samples required to learn these functions using a dense model. This lower bound implies that a forward pass on a dense model takes time $\Omega(d(2/\sqrt{k}\epsilon)^k)$, which is exponentially worse than the time taken by the sparse model. Altogether, we show that for the general class of Lipschitz functions, sparsely activated layers are as expressive as dense layers, while performing significantly fewer floating point operations (FLOPs) per example.

To support our theory, we perform experiments in Section 5 on approximating Lipschitz functions with sparse networks. We identify several synthetic datasets where models with data-dependent sparse layers outperform dense models of the same size. Moreover, we achieve these results with relatively small networks. This provides a few minimum working examples of when sparsity is beneficial. Our results complement the existing work that has already shown the success of very large MoE models on real-world NLP and vision tasks [14, 21, 33]. Our experiments also generalize the intuition from Figure 1a, where the model outputs a different constant value for each bucket. A puzzling empirical result was observed in [34] where a pseudo-random uniform hash function is used to route input tokens to experts. Intuitively, this might map very nearby or similar points to distinct experts, leading to a potentially non-smooth behavior. We show that on our synthetic, Lipschitz datasets, the LSH model outperforms uniform random hashing, which further corroborates our theory.

**Related Work.** Sparsely activated networks have had enormous empirical success [1, 5, 13, 25, 29, 33, 37, 40]. The Switch Transformer [14] is one of the first major, contemporary applications of the sparsely activated layers. Follow-up works such as Scaling Transformers [21] and other hash functions [34] aim to improve the sparse layers. These papers build upon seminal MoE works [19, 20, 22], and other uses of the MoE paradigm [27, 36]. There has been work on using statistical mechanics to analyze the generalization of MoE models [23]. However, to the best of our knowledge, there is no systematic theoretical study of modern sparsely activated networks.

The above work on dynamic sparsity builds on previous *static* sparsity efforts, e.g., weight quantization [28], dropout [38], and pruning (see the survey [18] and references). Static sparsity means that

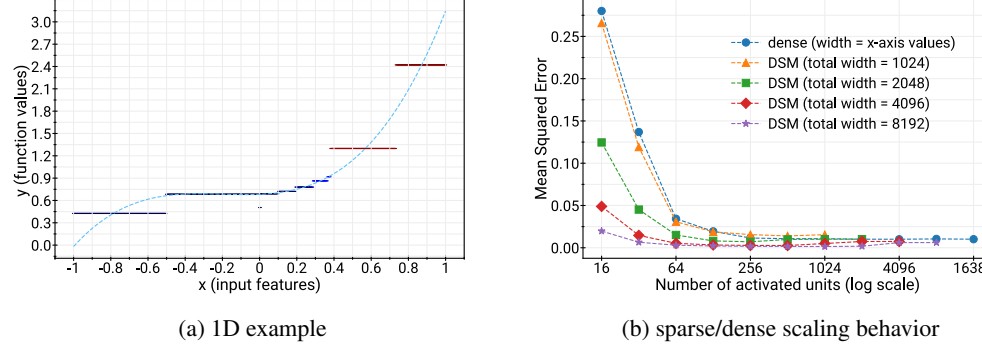

| (a) 1D example | (b) sparse/dense scaling behavior |

Figure 1: Learning polynomial functions with data-dependent sparse models: (a) Dashed curve is the target function graph, piecewise constant curve is the learned LSH model output, and different LSH buckets are indicated by different colors. Note neighboring points are hashed to the same bucket, where a simple constant can be learned to approximate target function values; (b) Scaling behavior of data-dependent sparse models (DSM) versus dense models, in which sparse models outperform dense models of the same size, see Section 2 and Section 5 for details.

the subnetwork activation does not change in a data-dependent way. The focus is on generalization and compression, instead of achieving fast inference time with a huge number of parameters.

Our work builds on locality sensitive hashing (LSH), a well-studied technique for approximate nearest neighbor search (see the survey [4] or the book [16] and references therein for LSH background). A seminal and popular LSH family uses random hyperplanes to partition the space [3, 7]. Another option is to use balls centered at lattice points, which works well for $\ell_2$ distance [2]. We use these ideas to design sparse networks. For uses of LSH in deep learning, sketch-based memory improves network capacity [15, 30]. Other empirical work uses LSH to improve dense network training time or memory [8, 9, 31]. Our work differs from the prior studies because we implement the LSH-based approach with sparsely activated networks, with the goal of reducing inference time and achieving low regression error. LSH-based sparsity may be most relevant for vision transformers [24, 33, 42], using distance metrics and routing functions tailored to images, or for time series analysis [41].

## 2 Preliminaries

Let $f : \mathbb{R}^d \to \mathbb{R}$ be a multivariate real-valued function that we want to learn with a neural network. We consider regression, and we aim to minimize the mean-squared error or the $\ell_\infty$ error. For a parameter $n \in \mathbb{N}^+$, a set $\Gamma$, a function $f : \Gamma \to \mathbb{R}$ and an estimator $\hat{f}_n : \Gamma \to \mathbb{R}$, we define $\|f - \hat{f}_n\|_\infty = \sup_{x \in \Gamma} |f(x) - \hat{f}_n(x)|$. For some results, we approximate $f$ on a subset $\mathcal{V} \subseteq \mathbb{R}^d$. For example, $\mathcal{V}$ may be the intersection of $[-1, 1]^d$ and a $k$-dimensional subspace. We say that $f$ is $L$-Lipschitz if $|f(x) - f(x')| \leq L \cdot \|x - x'\|_2$ for all $x, x' \in \mathbb{R}^d$. Let $[n] = \{1, 2, \ldots, n\}$.

### 2.1 Data-Dependent Sparse Model

A dense neural network $g$ with a fully connected final layer can be expressed as $g(x) = A \cdot \phi(x)$ where $A \in \mathbb{R}^{1 \times t}$ is a matrix and $\phi : \mathbb{R}^d \to \mathbb{R}^t$ is a function (e.g., $\phi$ captures the representation learned up until the final layer). Here, $t$ is the width of the final layer and $d$ is the input dimensionality.

Our focus is on the types of networks captured by what we call the *Data-Dependent Sparse Model (*DSM*)*. This is a network with a sparsely activated final layer. Formally, let $t$ be the width, and let $s \leq t$ be a sparsity parameter. Then, we consider functions $g$ of the form $g(x) = A^x \cdot \phi(x)$ where $A^x \in \mathbb{R}^{1 \times t}$ and $\phi : \mathbb{R}^d \to \mathbb{R}^t$. The crux of the model is the final layer. The sparsity comes from letting $A^x = A \circ \mathsf{mask}(x)$, where $\mathsf{mask}(x) \in \{0, 1\}^{1 \times t}$ is an $s$-sparse indicator vector, and "$\circ$" is the entry-wise product. The mask zeroes out certain positions, and $A$ contains the learned weights but no longer depends on $x$. Intuitively, the mask is the "routing function" for the sparse activations. To do

so, we define $\mathsf{mask}(x) \circ \phi(x)$ to mean that we zero out element of the vector $\phi(x)$. Then, we have

$$g(x) = (A \circ \mathsf{mask}(x))\phi(x) = A(\mathsf{mask}(x) \circ \phi(x)).$$

Under the above definitions, let $\mathsf{DSM}(d, s, t)$ be the set of functions $g = (A \circ \mathsf{mask}(x)) \cdot \phi(x)$. In what follows, we use $A^x$ as shorthand for $A \circ \mathsf{mask}(x)$.

In the DSM model, we wish to understand the effect of sparsity on how well the network can approximate certain functions. It is important to allow the final layer $A^x \in \mathbb{R}^{1 \times t}$ to be data-dependent. The values in $A^x$ depend on the target function $f$, and we learn this from training data.

Real-world transformers may have multiple sparse layers. To capture this, we can compose DSM functions. For example, two sparse layers comes from $g(x) = A \circ \mathsf{mask}_2(x) \circ \phi_2(\mathsf{mask}_1(x) \circ \phi_1(x))$. For simplicity, we focus on a single sparse layer in what follows, which suffices for our main results.

## 2.2 Hash-based Routing

Prior sparse models compute hash functions of the input vector $x$ to determine mask and the network activations [34]. Our main theorem uses this strategy, considering LSH families. LSH has the property that nearby points are more likely to end up in the same hash bucket than far away points. We can use LSH to define a general class of efficient regression functions. In Section 3, we prove that the DSM model captures popular sparse architectures and the following LSH model.

**LSH Model.** We review an LSH framework based on sign patterns. Let $h_1, \ldots, h_m : \mathbb{R}^d \to \{-1, 1\}$ be $m$ distinct hash functions. Partition the space into $2^m$ buckets based on the $m$ sign patterns $z_x = (h_1(x), \ldots, h_m(x))$ over all $x \in \mathbb{R}^d$. Then, for each $z \in \{-1, 1\}^m$, we can specify a function $\hat{g}_z(x)$, where the goal is for $g_z$ to approximate the target function $f$ in the part of space associated with $z$ (i.e., points $x$ that have pattern $z$ under $h_1, \ldots, h_m$). More generally, we can allow $s$ sets of such hash functions $(h_1^i, \ldots, h_m^i)$, and $s$ sets of these approximation functions $(\hat{g}_{z^1}^1, \ldots, \hat{g}_{z^s}^s)$ for $i = 1, \ldots, s$. On input $x$, we compute the sign patterns $z^1, \ldots, z^s$ and output $g(x) = \sum_{i=1}^s \alpha_i \hat{g}_{z^i}^i(x)$. Further, we can restrict each $\hat{g}_z^i$ to be a degree $\Delta \geq 0$ polynomial. For a fixed LSH family of possible hash functions, we let $\mathsf{LSH}(d, s, m, \Delta)$ denote this class of functions. In many cases, we only need $\hat{g}_z^i$ to be a constant function, i.e., $\Delta = 0$, and we shorten this as $\mathsf{LSH}(d, s, m, 0) := \mathsf{LSH}(d, s, m)$. Our goal is to understand the trade-offs of the "memory" parameter $m$ and the "sparsity" amount $s$ that we need to approximate large classes of $d$-dimensional functions (i.e., $L$-Lipschitz functions supported on a $k$-dimensional manifold).

**Euclidean** LSH **Model [11].** This is a popular LSH family for points in $\mathbb{R}^d$. In this case, each hash function outputs an integer (instead of $\pm 1$ above). Each bucket in this model is defined by a set of hyperplane inequalities. There are two parameters $(m', \epsilon)$ with $m' \in \mathbb{N}^+$ and $\epsilon \in (0, 1]$. We sample $m'$ random directions $a_1, \ldots, a_{m'}$ with each coordinate of each $a_i$ being an independent standard normal random variable. In addition, we sample $b_i \sim \mathrm{Unif}[0, \epsilon]$ independently for $i \in [m']$. For a point $x \in \mathbb{R}^d$, we compute an index into a bucket via a function $h_i : \mathbb{R}^d \to \mathbb{Z}$ defined as

$$h_i(x) = \left\lfloor \frac{a_i^\top x + b_i}{\epsilon} \right\rfloor.$$

Here, the index $i$ ranges over $i \in [m']$, leading to a vector of $D$ integers.

## 3 Simulating Models with DSM

We formally justify the $\mathsf{DSM}(d, s, t)$ model by simulating other models using it. We start with simple examples (interpolation and $k$ nearest neighbor ($k$-NN) regression), then move on to transformers and the LSH model. For the simple examples, we only need one hidden layer, where $\phi(x) = \sigma(Bx)$ for a matrix $B \in \mathbb{R}^{t \times d}$ and non-linearity $\sigma$.

**Interpolation.** We show how to compute $f$ at $t$ points $x_1, \ldots, x_t$. We only need sparsity $s = 1$ and $\sigma$ is simply the identity function. We set $A_i = f(x_i)/\langle b_i, x_i \rangle$, where $b_i$ is the $i$th row of $B$. Further, we let $\mathsf{mask}(x_i)$ have a one in the $i$th position and zeroes elsewhere. Then, we define $g(x_i) = (A \circ \mathsf{mask}(x_i))Bx_i = f(x_i)$, which computes $f$ at the $t$ input points.

**Sparse networks perform $k$-NN regression**.

We sketch how the DSM$(d, k, n)$ model can simulate $k$-NN with $g(x) = (A \circ \mathsf{mask}(x))\sigma(Bx)$, when the dataset contains vectors with unit $\ell_2$ norm. Let the rows of $B$ be a set of $n$ unit vectors $b_1, \ldots, b_n \in \mathbb{R}^d$. For the target function $f$, let $A = \frac{1}{k}(f(b_1), \ldots, f(b_n))$. Define $\sigma(Bx)$ to have ones in the top $k$ largest values in $Bx$ and zeroes elsewhere. For a unit vector $x$, these $k$ positions correspond to the $k$ largest inner products $\langle x, b_i \rangle$. Since $\|x - b_i\|_2 = 2 - 2\langle x, b_i \rangle$, the non-zero positions in $\sigma(Bx)$ encode the $k$ nearest neighbors of $x$ in $\{b_1, \ldots, b_n\}$. Thus, $g(x)$ computes the average of $f$ at these $k$ points, which is exactly $k$-NN regression. Moreover, only $k$ entries of $A$ are used for any input, since $\sigma(Bx)$ is $k$-sparse; however, computing $\sigma(Bx)$ takes $O(nd)$ time. While there is no computational advantage from the sparsity in this case, the fact that DSM can simulate $k$-NN indicates the power of the model.

### 3.1 Simulating transformer models with DSM

Real-world networks have many hidden layers and multiple sparse layers. For concreteness, we describe how to simulate a sparsely activated final layer. As mentioned above, we can compose functions in the DSM model to simulate multiple sparse layers.

**Switch Transformers [14].** The sparse activations in transformers depend on a *routing function* $R : \mathbb{R}^d \to \{1, \ldots, \ell\}$, where $R(x)$ specifies the subnetwork that is activated (for work on the best choice of $R$, see e.g., [34]). To put this under the DSM$(d, s, t)$ model, consider a set of trainable matrices $A_1, \ldots, A_\ell \in \mathbb{R}^{1 \times s}$, where the total width is $t = s \cdot \ell$ for integral $\ell \geq 1$. On input $x$, we think of $A^x$ as a $1 \times t$ matrix with $s$ non-zero entries equal to $A_{R(x)}$. In other words, $A$ is the concatenation of $A_1, \ldots, A_\ell$, and $\mathsf{mask}(x)$ is non-zero on the positions corresponding to $A_{R(x)}$.

**Scaling Transformers [21].** The key difference between Switch and Scaling transformers is that the latter imposes a block structure on the sparsity pattern. Let $t$ be the width, and let $s$ be the number of blocks (each of size $t' = t/s$). Scaling Transformers use only one activation in each of the $s$ blocks. In the DSM$(d, s, t)$ model, we capture this with $A^x$ as follows. Let $e_i \in \{0, 1\}^{t'}$ denote the standard basis vector (i.e., one-hot encoding of $i \in [t']$). The sparsity pattern is specified by indices $(i_1, \ldots, i_s)$. Then, $A^x = (\alpha_1 e_{i_1}, \ldots, \alpha_b e_{i_s})$ for scalars $\alpha_1, \ldots, \alpha_s \in \mathbb{R}$.

### 3.2 Simulating the LSH model using DSM

We explain how to represent the LSH model using the DSM model. The key aspect is that $A^x$ depends on the LSH buckets that contain $x$, where we have $s$ non-zero weights for the $s$ buckets that contain each input. This resembles the Scaling Transformer, where we use the LSH buckets to define routing function. In the LSH$(d, s, m, \Delta)$ model, there are $s$ sets of $m$ hash functions, leading to $s \cdot 2^m$ hash buckets. We use width $t = s \cdot 2^m$ for the DSM$(d, s, t)$ network. The entries of $A^x$ are in one-to-one mapping with the buckets, where only $s$ entries will be non-zero depending on the $s$ buckets that $x$ hashes to, that is, the values $(h_1^i(x), \ldots, h_m^i(x)) \in \{-1, 1\}^m$ for $i = 1, 2, \ldots, s$.

We now determine the values of these $s$ non-zero entries. We store a degree $\Delta$ polynomial $\hat{g}(x) : \mathbb{R}^d \to \mathbb{R}$ associated with each bucket. For our upper bounds, we only need degree $\Delta = 0$, but we mention the general case for completeness. If $\Delta = 0$, then $\hat{g}$ is simply a constant $\alpha$ depending on the bucket. An input $x$ hashes to $s$ buckets, associated with $s$ scalars $(\alpha_1, \ldots, \alpha_s)$. To form $A^x$, set $s$ entries to the $\alpha_i$ values, with positions corresponding to the buckets. For degree $\Delta \geq 1$, we store coefficients of the polynomials $\hat{g}$, leading to more model parameters. Section 4 contains details on using LSH to approximate Lipschitz functions with sparse networks.

**Computing and storing the LSH buckets.** Determining the non-zero positions in $A^x$ only requires $O(sm)$ hash computations, each taking $O(d)$ time with standard LSH families (e.g., hyperplane LSH). We often take $m$ to be a large constant. Thus, the total number of operations to compute a forward pass in the network $O(smd) \approx O(sd)$. The variable $m$ above determines the total number of distinct buckets we will have ($2^m$). For an $n$ point dataset, $m = O(\log n)$ is a realistic setting in theory. Therefore, $2^m = \mathrm{poly}(n)$ is often a reasonable size for the hash table and moreover the dense model requires width $O(2^m)$ as well. In practice, the number of experts is usually 32–128 [14, 33]. The hash function typically adds very few parameters. For example, to hash $d$ dimensional data into $2^m$ buckets, LSH requires $O(md)$ bits and takes $O(md)$ time as well. In summary, the LSH computation does not asymptotically increase the number of FLOPs for a forward pass in the network. We also later bound the number of non-empty LSH buckets. The memory to store the hash table is comparable to the number of samples required to learn the target function.

# 4 Data-Dependent Sparse Models are more Efficient than Dense Models

For a general class of Lipschitz functions, LSH-based learners yield similar $\ell_\infty$ error as dense neural networks while making inference significantly more efficient. It is a common belief in the machine learning community that although many of the datasets we encounter can appear to live in high-dimensional spaces, there is a low-dimensional manifold that contains the inputs. To model this, we assume in our theory that the inputs lie in a $k$-dimensional subspace (a linear manifold) of $\mathbb{R}^d$. Here $k \ll d$. Theorem 4.1 shows that the LSH model we propose can learn high-dimensional Lipschitz functions with a low $\ell_\infty$ error efficiently when the input comes from a uniform distribution on an unknown low-dimensional subspace. Theorem B.1 extends this result to when the input comes from an unknown manifold with a bounded curvature. We present Theorem 4.1 here and defer Theorem B.1 to Section B. All proofs are in the appendix.

**Theorem 4.1.** *For any $f : [-1, 1]^d \to \mathbb{R}$ that is $L$-Lipschitz, and for an input distribution $\mathcal{D}$ that is uniform on a $k$-dimensional subspace in $[-1, 1]^d$, an LSH-based learner can learn $f$ to $\epsilon$-uniform error with $O(kL\sqrt{d}^k \log(L\sqrt{d}/\epsilon)/\epsilon^k)$ samples using a hash table of size $O(L\sqrt{d}^k/\epsilon^k)$ with probability $\geq 0.8$. The total time required for a forward pass on a new test sample is $O(dk \log(L\sqrt{d}/\epsilon))$.*

The key idea behind this theorem is to use LSH to produce a good routing function. The locality of the points hashed to an LSH bucket gives us a way to control the approximation error. By using a large number of buckets, we can ensure their volume is small. Then, outputting a representative value suffices to locally approximate the target Lipschitz function (since its value changes slowly). The next lemma bounds the size of the sub-regions corresponding to each bucket, as well as bounding the number of non-empty buckets.

**Lemma 4.2.** *Consider a Euclidean* LSH *model in $d$ dimensions with $Ck$ hyperplanes and width parameter $\epsilon$ where $C$ is a large enough constant. Consider a region $\Gamma$ defined by the intersection of a $k$-dimensional subspace with $[-1, 1]^d$. We have that the* LSH *model defines a partitioning of $\Gamma$ into buckets. Let $c$ be a constant. Then, with probability $\geq 0.9$,*

*1. Projecting any bucket of the LSH onto $\Gamma$ corresponds to a sub-region with diameter $\leq \epsilon/c$.*

*2. At most $\left(\frac{2\sqrt{d}}{\epsilon}\right)^{O(k)}$ buckets have a non-empty intersection with $\Gamma$.*

The above construction assumes knowledge of the dimensionality of the input subspace $k$. Fortunately, any upper bound on $k$ would also suffice. The table size of the LSH model scales exponentially in $k$ but not $d$. Thus, an LSH-based learner adapts to the dimensionality of the input subspace.

Theorem 4.1 shows that sparsely activated models (using LSH of a certain table size) are powerful enough to approximate and learn Lipschitz functions. Next, we show a complementary nearly matching lower bound on the width required by dense model to approximate the same class of functions. We use an information theoretic argument.

**Theorem 4.3.** *Consider the problem of learning $L$-Lipschitz functions on $[-1, 1]^d$ to $\ell_\infty$ error $\epsilon$ when the inputs to the function are sampled from a uniform distribution over an unknown $k$-dimensional subspace of $\mathbb{R}^d \cap [-1, 1]^d$. A dense model of width $w$ with a random bottom layer requires*

$$w = \Omega\left(\frac{(\sqrt{d}L)^k}{(C\epsilon)^k}\right),$$

*for a constant $C > 0$ to learn the above class of functions. The number of samples required is*

$$\Omega\left(\frac{k(\sqrt{d}L)^k \log(2\sqrt{d}L/C\epsilon)}{(C\epsilon)^k}\right).$$

Our approach for this theorem is to use a counting argument. We first bound the number of distinct functions, which is exponential in the number of parameters (measured in bits). We then construct a large family of target functions that are pairwise far apart from each other. Hence, if we learn the wrong function, we incur a large error. Our function class must be large enough to represent any possible target function, and this gives a lower bound on the size of the approximating network.

Theorem 4.3 shows a large gap in the time complexity of inference using a dense model and an LSH model. The inference times taken by a dense model vs. a sparse model differ exponentially in $1/\epsilon$.

$$\textbf{Sparse: } O\left(dk\log(1/\epsilon)\right) \qquad \text{vs.} \qquad \textbf{Dense: } \Omega\left(d\left(\frac{2\sqrt{d}}{C\epsilon}\right)^k\right) \qquad (1)$$

Overall, the above theorems show that LSH-based sparsely activated networks can approximate Lipschitz functions on a $k$-dimensional subspace. The size and sample complexity match between sparse and dense models, but the sparse models are exponentially more efficient for inference.

## 5   Experiments

To empirically verify our theoretical findings, we align our experiments with our proposed models and compare dense models, data-dependent sparse models (DSM), LSH models, and sparse models with random hash based sparse layers. While DSM and LSH models are analyzed in the previous section, random hash based layers introduce sparsity with a random hash function, as an alternative of learnable routing modules and LSH in MoE models [34]. Our goal is to show that the DSM and LSH models achieve small MSE while using much fewer activated units than dense models. We also report similar observations on the CIFAR-10 dataset.

### 5.1   Experimental set-up

Dense, DSM and random hash sparse models contain a random-initialized, non-trainable bottom layer, (a Top-K layer for DSM and a random hash layer for random hash models to enforce sparsity), and a trainable top layer, with varying number of hidden units and sparsity levels. LSH models have non-trainable hyperplane coefficients for hashing and a trainable scalar in each bucket (the scalar determines the output of the network for points in that bucket). We compare dense models and three sparse models (DSM, LSH, and random).

We evaluate with synthetic data from two random, Lipschitz target functions that are commonly used basis functions for arbitrary continuous functions. These random functions allow us to empirically evaluate the construction from Theorem 4.1, while comparing different routing functions.

**Random polynomial.** $p(x)$ of degree $d$ for $x \in \mathbb{R}^n$ with sum of coefficient absolute values $< 1/(d)$. As a result, we ensure that the function is 1-Lipschitz over $[-1, 1]^n$.

**Random hypercube function.** $f : [-1, 1]^n \to \mathbb{R}$ which interpolates the indicator functions at each corner with random $\{-1, 1\}$ value at each corner. Concretely, the function is defined as follows: for each corner point $y \in \{-1, 1\}^n$, its indicator function is $I_y(x) = \prod_{i=1}^{n} \frac{1+y_i x_i}{2}$. Sample random values $v_y \in \{-1, 1\}$ with probability $(0, 5, 0.5)$ independently for each $y \in \{-1, 1\}^n$, the random hypercube polynomial function is $f(x) = \sum_{y \in \{-1,1\}^n} v_y I_y(x)$.

**Random 3-layer neural net function.** $f : [-1, 1]^n \to \mathbb{R}$ is a randomly initialized 3-layer neural network with fully connected layers.

For the random polynomial (random hypercube) functions, we use $n = 8$ and $d = 4$ in the this section and provide more results for other parameter settings in the appendix (especially the case when the target polynomial functions concentrate on a low dimension subspace). We also present results on a real dataset, CIFAR-10, which corroborates our findings from the synthetic data experiments.

### 5.2   Results

**MSE for random functions.** Figures 2 and 3 show the scaling behavior of the MSE for DSM and LSH models on a random polynomial function and hypercube function (lower is better). Sparsity helps for the DSM and LSH models, both achieving better quality than dense models using the same number of activated units. In Figure 4, we compare the DSM/LSH models with the random hash sparse models, and we see random hash sparse models underperform dense models, suggesting data-dependent sparsity mechanisms (such as DSM/LSH) are effective ways to utilize sparsity. Table 2 shows the scaling behavior of DSM and dense models for the randomly initialized 3-layer NN target

function. We see that at the same total width a sparser DSM model can perform as well as a dense model.

**FLOPs.** To further qualify the efficiency gain of sparse models, we compare the MSE at the same FLOPs for sparse/dense models in Table 1. The first column is the # FLOPs for the dense model; models in the 3rd and 4th columns use same # FLOPs but have more parameters (only 50% or 25% active). DSM uses only 18k FLOPs and gets smaller MSE than dense model with 73k FLOPs.

| FLOPs | eval MSE (dense) | eval MSE (DSM 50% sparsity) | eval MSE (DSM 25% sparsity) |
|-------|------------------|------------------------------|------------------------------|
| 18432 | 0.01015 | 0.01014 | **0.009655** |
| 36864 | 0.01009 | 0.007438 | **0.005054** |
| 73728 | 0.01046 | 0.006115 | **0.001799** |

Table 1: FLOPs and evaluation Mean Squared Error (eval MSE).

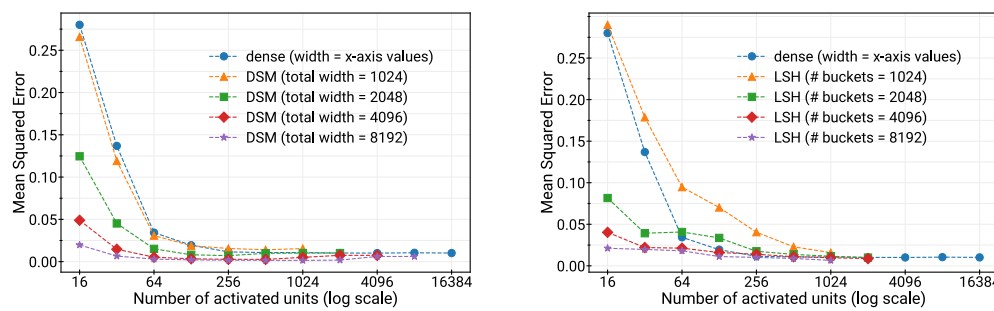

Figure 2: Scaling behavior of DSM and LSH models compared with dense models for a degree $4$ random polynomial with input dimension $8$: (a) DSM outperforms dense model at the same number of activated units and quality improves as total width increases; (b) LSH model outperforms dense model when number of buckets is large ($\geq 2048$) and quality improves as number of buckets increase.

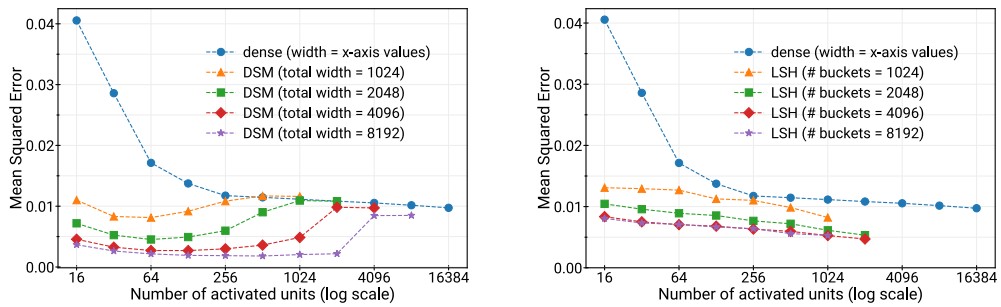

Figure 3: Scaling behavior of DSM and LSH models compared with dense models for a random hypercube function with input dimension $8$. Both DSM and LSH models outperform corresponding dense models with the same number of activated units (note: # activ. units $\leq$ width).

| Model \Width | 1024 | 2048 | 4096 |
|--------------|------|------|------|
| Dense | 4.32e-5 | 1.76e-5 | 7.4e-6 |
| DSM with sparsity 32 | 1.3e-4 | 7e-5 | 2.3e-5 |
| DSM with sparsity 64 | 1.1e-4 | 3.1e-5 | 1.9e-5 |
| DSM with sparsity 128 | 1e-4 | 4.2e-5 | 1.5e-5 |
| DSM with sparsity 256 | 8.7e-5 | 3.7e-5 | 1.7e-5 |
| DSM with sparsity 512 | 5.8e-5 | 3.8e-5 | 1.9e-5 |
| DSM with sparsity 1024 | 6e-5 | 2.1e-5 | 2.2e-5 |

Table 2: MSE for random 3-layer NN target function.

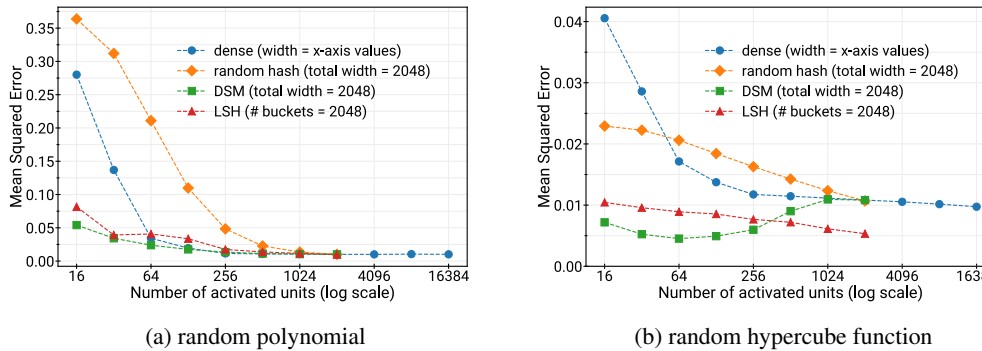

|                          |                          |
| :----------------------: | :----------------------: |
| (a) random polynomial    | (b) random hypercube function |

Figure 4: Scaling behavior of dense, random hash, DSM, and LSH models. DSM and LSH models outperform dense models, while random hash models underperform dense models with the same number of activated units.

**CIFAR-10.** We also compare the scaling behavior of DSM and dense models on CIFAR-10 [26]. The first baseline model is a CNN with 3 convolutional layers (followed by max-pooling), a dense layer with varying number of units, and a final dense layer that computes the logits (referred as CNN + dense). For the data-dependent sparse models, we use the same architecture, except we change the penultimate dense layer with a data-dependent sparse layer (referred as CNN + DSM). Both models are trained with ADAM optimizer for 50 epochs and evaluated on the test dataset for model accuracy with no data augmentation; see Figure 5 and Table 3 for the accuracy versus number of activated units. As with the synthetic datasets, DSMs outperform dense models at the same number of activated units. Next we also compare the scaling behavior of DSM and dense variants of Vision Transformer models on CIFAR-10. These results are presented in Table 4.

### 5.3 Discussion

Our experimental results (e.g., Figures 2, 3, and 4) show that sparsely activated models can efficiently approximate both random polynomials and random hypercube functions. Intuitively, the DSM and LSH models employ the sparsity as a way to partition the space into nearby input points. Then, because the target function is Lipschitz, it is easy to provide to local approximation tailored to the specific sub-region of input space. On the other hand, the uniform random hash function performs poorly for these tasks precisely because it does not capture the local smoothness of the target function.

On CIFAR-10, we also see that the DSM model performs comparably or better than the dense network. In particular, Figure 5 shows that the "CNN + DSM" approach with total width 1024 improves upon or matches the dense model. In this case, the sparse activation allows the network to classify well while using only a fraction of the number of FLOPs. In Table 3, we see that DSM model outperforms the dense model when we control for the number of activated units in the comparison.

**Limitations.** Our experiments focus on small datasets and 2-layer networks as a way to align with our theoretical results. Prior work on sparsely activated networks has shown success for large-scale NLP and vision tasks. Our experiments complement previous results and justify the DSM and LSH models by showing their ability to approximate Lipschitz functions (consistent with our theorems). It would be good to further evaluate the DSM and LSH models for large-scale tasks, for example, by using them as inspiration for routing functions in MoE vision transformers, such as V-MoE [33]. It would be interesting to evaluate on larger datasets, such as ImageNet as well. We experimented with a handful of hyperparameter settings, but a full search may lead to different relative behavior between the models (similarly, we only evaluated a few parameter settings for the random functions and the synthetic data generation, which is far from exhaustive).

## 6 Conclusion

We provided the first systematic theoretical treatment of modern sparsely activated networks. To do so, we introduced the DSM model, which captures the sparsity in Mixture of Experts models, such as

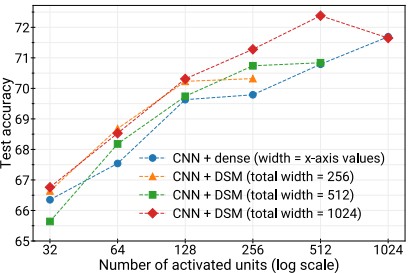

Figure 5: Scaling behavior of DSM compared with dense CNN models on the CIFAR-10 dataset. Similar to Figure 4a, DSM outperforms dense models at the same number of activated units.

| Model \ # activated units | 256 | 512 |
|---|---|---|
| Dense | 69.79 | 70.79 |
| DSM (50% sparse) | **70.74** | 71.33 |
| DSM (25% sparse) | 69.8 | **71.68** |

Table 3: CIFAR-10 test accuracy for dense/DSM CNN models with the same number of activated units. While not strictly monotonic, wider and sparser models outperform narrow and dense ones.

| Model | Hidden Dimension | Number of Activated Units | Sparsity | Top-1 Test Accuracy |
|---|---|---|---|---|
| Dense | 768 | 768 | 0% | 79.8% |
| DSM-512 | 1024 | 512 | 50% | 79.9% |
| DSM-384 | 1152 | 384 | 66.7% | 79.2% |
| DSM-128 | 1408 | 128 | 90.9% | 80.1% |

Table 4: CIFAR-10 test accurace for dense/DSM Vision Transformer (ViT) models with the same number of activated units.

**Switch and Scaling Transformers.** We showed that DSM can simulate popular architectures as well as LSH-based networks. Then, we exhibited new constructions of sparse networks. Our use of LSH to build these networks offers a theoretical grounding for sparse networks. We complemented our theory with experiments, showing that sparse networks can approximate various functions.

For future work, it would be interesting to implement LSH-based networks in transformers for language/vision tasks. A related question is to determine the best way to interpolate in each LSH bucket (e.g., a higher degree polynomial may work better). Another question is whether a dense model is more powerful than a sparse model with the same number of total trainable parameters. Theorem 4.1 only says that a sparse model with similar number of parameters as a dense model can more efficiently (fewer FLOPs) represent Lipschitz functions. This does not say *all* functions expressible by a dense model are also expressible by a sparse model. This is non-trivial question as $A^x$ depends on the input (i.e., $\mathrm{DSM}(d, t, t)$ may be more expressive than the dense model with width $t$). We expect that dense networks can be trained to perform at least as well as sparse networks, *assuming the width is large enough*. The dense networks should optimize the weights in the last layer to approximate the function, but they may not favor certain neurons depending on the input.

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
