# A   Proofs of Main Upper and Lower Bounds

## A.1   Proof of the Lipschitz Upper Bound

*Proof of Theorem 4.1.* We use the Euclidean-LSH construction of Lemma 4.2 with parameter $\epsilon/L$. In any sub-region of the $k$-dimensional subspace that has a small diameter, the Lipschitz nature of the function together with Lemma 4.2 will imply that we can approximate it by just a constant and incur only $\epsilon$ error in $\ell_\infty$. In particular, given a point $x_1$ belonging to an LSH bucket, we can set $\hat{f}(x) = f(x_1)$ everywhere in that bucket. For any $x_2$ also mapping to the same bucket, from Lemma 4.2, we have that $\|x_1 - x_2\|_2 \le \epsilon/L$. Since $f$ is L-Lipschitz,

$$|\hat{f}(x_2) - f(x_2)| = |f(x_1) - f(x_2)| \le L\|x_1 - x_2\|_2 \le \epsilon. \tag{2}$$

Next we look at how many samples we need to obtain the guarantee $\|\hat{f} - f\|_\infty \le \epsilon$. A rare scenario that we have to deal with for the sake of completeness is when there exist buckets of such small volume that no training data point has mapped to them and consequently we don't learn any values in those buckets. At test time, if we encounter this rare scenario of mapping to a bucket with no value learnt in it, we simply run an approximate nearest neighbor search among the train points. For our prediction, we use the bucket value associated with the bucket that the approximate nearest neighbor maps to. To control the error when doing such a procedure, we take enough samples to approximately form an $\epsilon/\Gamma L$ cover of $\Gamma$ for a large enough constant $\Gamma$. The size of an $\epsilon/\Gamma L$ cover of $\Gamma$ is $O((2\Gamma L\sqrt{d}/\epsilon)^k)$. This implies that, via a coupon collector argument, when the input distribution is uniform over the region $\Gamma$, $O(k(2\Gamma L\sqrt{d}/\epsilon)^k \log(2\Gamma L\sqrt{d}/\epsilon))$ samples will ensure that with very high probability, for every test point $x$ there exists a train example $x_i$ such that $\|x - x_i\|_2 \le 2\epsilon/\Gamma L$. The test error is $|f(x) - \hat{f}(x_i)| \le |f(x) - f(x_i)| + |f(x_i) - \hat{f}(x_i)| = O(\epsilon)$. Computing the exact nearest neighbor is a slow process. Instead we can compute the approximate nearest neighbor using LSH very quickly. We lose another $O(\epsilon)$ error due to this approximation. Choosing $\Gamma$ appropriately we can make the final error bound exactly $\epsilon$. This leads to our stated sample complexity bound.

This implies that $\|f - \hat{f}\|_\infty \le \epsilon$. Hence using an Euclidean LSH with $O(k)$ hyperplanes we can learn an $\epsilon$-approximation to $f$. The time to compute $\hat{f}(x)$ for a new example is the time required to compute the bucket id where it maps to. Since there are $k$ hyperplanes and our input is $d$-dimensional, computing the projections of $x$ on the $k$ hyperplanes takes $O(dk)$ time. Then we need to perform a division by the width parameter $\epsilon/L$. This takes time equal to the number of bits required to represent $L/\epsilon$. Hence the total time taken would be $O(dk\log(L/\epsilon))$.  $\square$

The above theorem uses a lemma about Euclidean LSH, which we prove next.

*Proof of Lemma 4.2.* Let $K = Ck$. Let the random hyperplanes chosen by the Euclidean LSH be $a_1, \ldots, a_K$. Let the width parameter used by the LSH be $\epsilon_{LSH}$. The value of $\epsilon_{LSH}$ we choose will be determined later. Since the distribution of entries is spherically symmetric, the projection of the vectors onto the $k$-dimensional subspace will also form a Euclidean-LSH model. Henceforth in our analysis we can assume that all our inputs are projected onto the $k$-dimensional space $\Gamma$ and that we are performing LSH in a $k$-dimensional space instead of a $d$-dimensional one. Let $A = [a_1, \ldots, a_K]^\top$ be the matrix whose columns are the vectors perpendicular to the hyperplanes chosen by the LSH. Note that $A \in \mathbb{R}^{K \times k}$. Then we have, from tail properties of the smallest singular value distribution of Gaussian random matrices (e.g. see [10, 35]), for a large enough constant $c$,

$$\Pr[\sigma_{\min}(A) \ge c\sqrt{k}] \ge 9/10. \tag{3}$$

For two points $x_1, x_2 \in \Gamma$ to map to the same LSH bucket, $\|A(x_1 - x_2)\|_\infty \le \epsilon_{LSH}$. This implies that $\|A(x_1 - x_2)\|_2 \le \epsilon_{LSH}\sqrt{k}$. Together with (3), this implies that $\|x_1 - x_2\|_2 \le \epsilon_{LSH}/c$ with probability $\ge 9/10$. At the same time, since we $x_1, x_2 \in [-1, 1]^d$, the maximum distance along any direction is at most the length of any diagonal, which is $2\sqrt{d}$. Moreover, along any hyperplane direction sampled by the LSH, we grid using a width $\epsilon_{LSH}$. Since the total number of hyperplanes is $Ck$ the maximum number of LSH buckets possible is $\left(\frac{2\sqrt{d}}{\epsilon_{LSH}}\right)^{O(k)}$. We set $\epsilon_{LSH} = c\epsilon$. Then, the diameter of any bucket $\le \epsilon_{LSH}/c = \epsilon$. The upper bound on the maximum number of LSH buckets required to cover the region $\Gamma$ also follows.  $\square$

## A.2 Proof of the Dense Lower Bound

*Proof of Theorem 4.3.* Assuming $B$ bits per parameter, in our dense layer model we have $2^{Bw}$ distinct possible configurations. We lower bound the width $w$ by constructing a class of functions $\mathcal{F}$ defined on a $k$-dimensional subspace within $[-1,1]^d$ such that three properties simultaneously hold:

1. each $f \in \mathcal{F}$ is $L$-Lipschitz,

2. the number of functions in $\mathcal{F}$ is at least $\Omega(2^{(2\sqrt{d}L/C\epsilon)^k})$

3. for $f_1 \neq f_2 \in \mathcal{F}$, we have $\|f_1 - f_2\|_\infty > \epsilon$.

These three properties together will imply that $w \geq \frac{1}{B}(2\sqrt{d}/C\epsilon)^k$ as otherwise by there would have to be two functions $f_1 \neq f_2 \in \mathcal{F}$ that are approximated simultaneously by the same dense network, which is impossible since $\|f_1 - f_2\|_\infty > \epsilon$. We construct $\mathcal{F}$ as follows. Given the $d$-dimensional cube $[-1,1]^d$, we pick a subset of $k$ diagonals of the cube such that they are linearly independent. We consider the $k$-dimensional region defined by the intersection of the subspace generated by these diagonals and the cube $[-1,1]^d$. Denote the region we obtain by $G$. Let $e_1, \ldots, e_k$ form an orthonormal basis for the subspace $G$ lies in. We grid $G$ into $k$-dimensional cubes of side length $2\epsilon/L$ aligned along its bases $\{e_i\}_{i=1}^k$. For the center of every cube we pick a random assignment from $\{+\epsilon, -\epsilon\}$. Then we interpolate the function everywhere in $G$ such that (i) it satisfies the assigned values at the centers of the cubes and (ii) its value decreases linearly to 0 with radial distance from the center. That is, given the set of cube centers $V$

$$f(x) = \sum_{v \in V} \max(0, f(v) - L\operatorname{sgn}(f(v))\|x - v\|_2)$$

To understand the Lipschitz properties of such an interpolation, note that the slope at any given point in $G$ is either 0 or $L$, which bounds the Lipschitz constant by $L$. The total number of cubes that lie within $G$ is at least $(\sqrt{d}L/C\epsilon)^k$ for some constant $C$ and hence $\mathcal{F}$ contains a total of $(2)^{(\sqrt{d}L/C\epsilon)^k}$ functions. Moreover, given any $f_1, f_2 \in \mathcal{F}$ such that $f_1 \neq f_2$, there exists a cube center where their values differ by $2\epsilon$ giving us the third desired property as well. Consequently, we get that to attain $\epsilon$-uniform error successfully on $\mathcal{F}$ we need

$$2^{Bw} \geq 2^{(\sqrt{d}L/(C\epsilon))^k},$$

which implies that $w = \Omega((\sqrt{d}L)^k/(C\epsilon)^k)$. $\qquad\square$

## B LSH Models Can Also Learn Lipschitz Functions on $k$-Manifolds

A $k$-dimensional manifold (referred to as a $k$-manifold) can loosely be thought of as a $k$-dimensional surface living in a higher dimensional space. For example the surface of a sphere in 3-dimensions is a 2-dimensional manifold. We consider $k$-manifolds in $\mathbb{R}^d$ that are homeomorphic to a $k$-dimensional subspace in $\mathbb{R}^d$. We assume that our $k$-dimensional manifold $M_k$ is given by a transform $f : R^k \to R^k$ applied on $k$-dimensional subspace of $\mathbb{R}^d$ $L_k$. To control the amount of distortion that can occur when going from $L_k$ to $M_k$, the Jacobian of $f$ is assumed to have a constant condition number for all $x \in L_k$. We now state our main upper bound for manifolds, showing that LSH models can adapt and perform well even with non-linear manifolds of a bounded distortion from a linear subspace.

**Theorem B.1.** *For any $f : [-1,1]^d \to \mathbb{R}$ that is 1-Lipschitz, and for an input distribution $\mathcal{D}$ that is uniform on a $k$-manifold in $[-1,1]^d$, an* LSH *model can learn $f$ to $\epsilon$-uniform error with $O(k\sqrt{dk}^k \log(\sqrt{dk}/\epsilon)/\epsilon^k)$ samples using a hash table of size $O(\sqrt{dk}^k/\epsilon^k)$ with probability $\geq 0.8$. The total time required for a forward pass on a new test sample is $O(dk\log(1/\epsilon))$.*

*Proof.* The main idea of the proof is to follow similar arguments from Theorem 4.1 on the subspace $L_k$ and try to bound the amount of distortion the arguments face when mapped to the manifold $M_k$. Since we are no longer dealing with a subspace (linear manifold), the argument that an LSH in $d$-dimensions can be viewed as an equivalent LSH in $k$-dimensions does not hold. We use Euclidean-LSH models with $O(d)$ hyperplanes. Furthermore, we will use multiple LSH models each defined

using $O(d)$ hyperplanes. The main challenge in the proof is to show that the total number of buckets used in approximating $f$ do not grow exponentially in $d$, which is a possibility now as we use $O(d)$ hyperplanes.

**Lemma B.2.** *For any $x \in \mathbb{R}^d$, a $d$-dimensional sphere of radius $O(\epsilon/d)$ centered at $x$ is fully contained in the bucket where the center of the sphere maps to with probability $\geq 0.9$.*

*Proof.* Along any hyperplane direction the gap between parallel hyperplanes is $\epsilon$. Since any point is randomly shifted before being mapped to a bucket we get that with probability $1 - O(1/d)$, $x$ is more than $\Omega(1/d)$ away from each of the two parallel hyperplanes on either side. So with probability $(1 - O(1/d))^{O(d)} = \Omega(1)$ the entire sphere is contained inside the LSH bucket $x$ maps to. □

**Lemma B.3.** *Using $O(k \log d)$ Euclidean-LSH functions, we get that every $x \in L_k$, there exists a bucket in at least one of the $O(k \log d)$ buckets $x$ gets mapped to such that the entire $k$-dimensional sphere of radius $O(\epsilon/d)$ centered at $x$ is contained within the bucket.*

*Proof.* We use a covering number argument. The maximum volume of a $k$-dimensional subspace within $[-1, 1]^d$ is $(2\sqrt{d})^k$. We cover this entire volume using spheres of radius $\epsilon/d$. The total number of spheres required to do this are $O((2d\sqrt{d})^k/\epsilon^k)$. We now do a union bound over all the sphere centers in our cover above. For a single sphere, the probability that it does not go intact into a bucket in any of the $O(k \log d)$ LSH functions is $d^{-\Omega(k)}$. By a union bound, we can bound the probability that there exists a sphere center that does not go into a bucket to be $d^{-\Omega(k)}$. Hence the Lemma statement holds with exceedingly large probability of $1 - d^{\Omega(k)}$. □

Now, we only include buckets with volume at least $(\Omega(\epsilon/(d\sqrt{k})))^k$. We can do this procedure using approximate support estimation algorithms [39]. This takes time and sample complexity $S/\log S$ where $S$ is the size of the support. With constant probability all points in $L_k$ are mapped to some such high volume bucket in at least one of the LSH functions. The total number of buckets with this minimum volume is at most $(O((d^2\sqrt{k})/\epsilon))^k$, which is also an upper bound on the sample complexity and running time of the support estimation procedure. Now, we lift all the above results when we go to $M_k$ from $L_k$. Since the Jacobian of the manifold map $f$ has a constant condition number, its determinant is at most $\exp(k)$; so the volume of any region in $L_k$ changes by at most an $\exp(\pm O(k))$ multiplicative factor when it goes to $M_k$. So all volume arguments in the previous proofs hold with multiplicative factors $\exp(\pm O(k))$. This concludes our proof. □

## C   Lower Bound for Analytic Functions

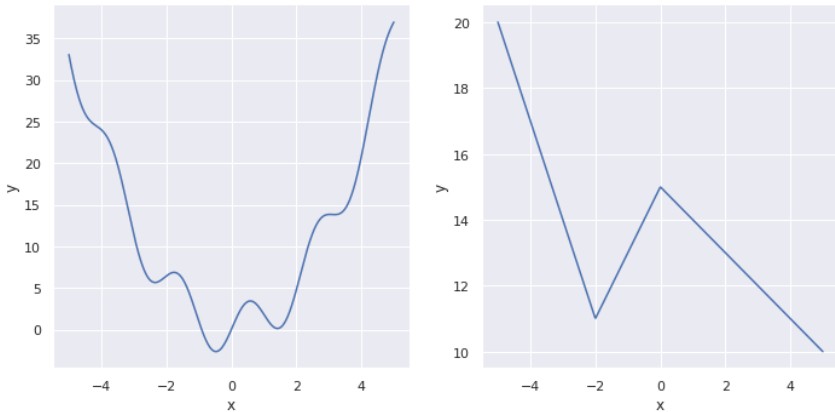

Figure 6: On the left we have an example of an analytic function. The function on the right is not analytic. Both functions have a bounded Lipschitz constant.

The functions described in the lower bound presented earlier are continuous but not differentiable everywhere as they are piecewise linear functions. In Theorem C.1 we show that we can make the lower bound stronger by providing a construction of $L$-Lipschitz analytic functions (which are differentiable everywhere). See Figure 6 for an example of analytic vs non-analytic functions.

**Theorem C.1.** *A dense model of width $w$ with a random bottom layer requires*

$$w = \Omega\left(\frac{2^{k^2/2}(LC_1)^k}{(\sqrt{k}\pi\epsilon)^k}\right),$$

*where $C_1$ is a large enough constant, to learn $L$-Lipschitz analytic functions on $[-1,1]^d$ to $\ell_\infty$ error $\epsilon$ when the inputs are sampled uniformly over a unknown $k$-dimensional subspace of $\mathbb{R}^d \cap [-1,1]^d$. Moreover, the number of samples required to learn the above class of functions is*

$$\Omega\left(w \log w\right),$$

*where $w = \Omega\left(\frac{2^{k^2/2}(LC_1)^k}{(\sqrt{k}\pi\epsilon)^k}\right)$.*

*Proof of Theorem C.1.* We construct a family $\mathcal{F}$ of analytic functions that are $L$-Lipschitz and are described using the Fourier basis functions. Each $f \in \mathcal{F}$ will be of the form

$$f(x) = \sum_{n_1=0}^{\infty} \sum_{n_2=0}^{\infty} \cdots \sum_{n_k=0}^{\infty} a_{n_1 n_2 \ldots n_k} \exp\left(i\pi n^\top x\right),$$

for $x \in [-1,1]^k$. We pick a small value of $0 < \epsilon_1 < 1$. We assume $1/\epsilon_1$ is an integer for convenience. If it is not, we can simply take $\lceil 1/\epsilon_1 \rceil$ instead. For a set of integers $(n_1, n_2, \ldots, n_k) \in [1/\epsilon_1]^k$, let $\eta_{n_1 n_2 \ldots n_k} \in \{\pm\}$. We use $\eta_n$ as a shorthand when it is not ambiguous. The family $\mathcal{F}$ is defined as the set of functions $f$ below

$$f(x) = \sum_{n_1,\ldots,n_k=0}^{1/\epsilon_1} \eta_n \epsilon_1^\alpha \left(\exp(i\pi n^\top x) + \exp(i\pi n^\top x)\right), \tag{4}$$

where each $\eta_n$ is chosen to be either $\pm L/(C\sqrt{k}\pi)$ for a large enough constant $C$ and $\alpha$ will be determined later. There are $(1/\epsilon_1)^k$ Fourier bases in each $f$ and the coefficient of each is set to be $\pm L\epsilon_1^\alpha/(C\sqrt{k}\pi)$. Hence we have

$$|\mathcal{F}| = 2^{\left((1/\epsilon_1)^k\right)}. \tag{5}$$

Next we argue that a larger than $0.9$ fraction of the functions in $\mathcal{F}$ are $L$-Lipschitz. We have,

$$\nabla f(x) = \sum_{n_1,\ldots,n_k=0}^{1/\epsilon_1} \eta_n \epsilon_1^\alpha i\pi (\exp(i\pi n^\top x) - \exp(i\pi n^\top x))n$$

$$= \sum_{n_1,\ldots,n_k=0}^{1/\epsilon_1} -2\eta_n \pi \sin(\pi n^\top x)\epsilon_1^\alpha n \tag{6}$$

$$\implies \mathbb{E}\left[\nabla f(x)\right] = 0, \tag{7}$$

where the last expectation is over the uniform measure over functions in $\mathcal{F}$. To get a bound on $\|\nabla f(x)\|_2$ we bound each $(\nabla f(x))_i$ with high probability. Each $(\nabla f(x))_i$ is a sum of $(1/\epsilon_1)^k$ independent random variables, namely $\eta_n$. We saw above that $\mathbb{E}[(\nabla f(x))_i] = 0$. To bound $|(\nabla f(x))_i|$ with high probability we will use McDiarmid's inequality. An upper bound on how much the value of $(\nabla f(x))_i$ can change when any one $\eta_n$ flips in value is computed as $4\epsilon_1^{(\alpha-1)}L/C\sqrt{k}$. Then, an application of McDiarmid's concentration inequality gives us that,

$$\Pr\left[|(\nabla f(x))_i| > t\right] \leq 2\exp\left(\frac{-t^2 k C^2 \epsilon_1^{(k+2-2\alpha)}}{16L^2}\right),$$

$$\implies |(\nabla f(x))_i| \leq \frac{L}{\sqrt{k}\epsilon_1^{(k+2-2\alpha)/2}} \tag{8}$$

with probability $\geq 0.9$ for a large enough constant $C$. This implies that

$$\|\nabla f(x)\|_2 \leq \frac{L}{\epsilon_1^{(k+2-2\alpha)/2}} \tag{9}$$

with probability $\geq 0.9$ for a randomly sampled $f \in \mathcal{F}$. Now, let $\eta_f$ denote the vector of $\eta_n$ values in sequence for any $f$. Using McDiarmid's (or Hoeffding's) concentration bound again, we also get that, with probability $\geq 0.9$, the Hamming distance between $\eta_{f_1}$ and $\eta_{f_2}$ for two $f$ randomly sampled from $\mathcal{F}$ is at least $c(1/\epsilon_1)^k$ for a small enough constant $c < 1$. This implies that for randomly sampled $f_1, f_2$,

$$f_1(x) - f_2(x)$$
$$= \sum_{n_1,\dots,n_k=0}^{1/\epsilon_1} 2\eta_n' \epsilon_1^\alpha \left( \exp(i\pi n^\top x) + \exp(i\pi n^\top x) \right), \tag{10}$$

where $\eta_n'$ is non-zero for at least $c(1/\epsilon_1)^k$ of the terms from the above argument about the Hamming distance. Parseval's identity then implies that

$$\frac{1}{2^k} \int_{-1}^{1} \dots \int_{-1}^{1} (f_1(x) - f_2(x))^2 dx_1 \dots dx_k$$
$$\geq 4L^2 \epsilon_1^{2\alpha} c \frac{1}{\epsilon_1^k C^2 k \pi^2}$$
$$\implies \|f_1 - f_2\|_\infty \geq \frac{2^{(k/2+1)} L \sqrt{c} \epsilon_1^{(\alpha-k/2)}}{C\sqrt{k}\pi}. \tag{11}$$

Finally we note that by union bound, at least a $0.8$ fraction of the functions in $\mathcal{F}$ satisfy both our Lipschitzness property (9) and (11) simultaneously. Setting $\alpha = k/2 + 1$ and $\epsilon_1 = \frac{C\sqrt{k}\pi\epsilon}{L\sqrt{c}2^{k/2}}$ we get that to achieve a strictly smaller error than $2\epsilon$ in the $\|.\|_\infty$ sense, one requires a dense model with a width of

$$\Omega\left( 2^{k^2/2} \left( \frac{LC_1}{\sqrt{k}\pi\epsilon} \right)^k \right).$$

$\square$

# D  Experiment details

## D.1  Details of learning random functions

In Section 5, we demonstrated experiments for randomly generated polynomial/hypercube functions. Here we present the details for the experiment settings.

**Random function generation.** For the random polynomial functions, we randomly generate coefficients of the monomials by sampling from a uniform distribution $\mathcal{U}([-1, 1])$ and scale the coefficients so that their absolute values sum up to $1.0$ (this is to ensure the Lipschitz constant of the generated function is bounded by a constant independent of dimension and degree of the polynomial). For the random hypercube function, we sample values of the function at each corner independently from a uniform distribution on $-1, 1$, and interpolate using the indicator functions.

**Train/Test dataset generation.** For a given target function $f$ (polynomial or hypercube), we sample independently from $\mathcal{U}([-1, 1]^n)$ (where $n$ is the input dimension) to generate the input features $x$ and compute target value $y = f(x)$. The train dataset contains $2^{16}$ $(x, y)$ pairs and the test dataset contains $2^{14}$ $(x, y)$ pairs.

**Training setting.** All the models in Section 5 are trained for 50 epochs using the RMSProp [17] optimizer with a learning rate of $10^{-5}$. For the one dimension example in Section 1, the model is trained for 200 epochs using the RMSProp optimizer with a learning rate of $5 \times 10^{-6}$.

**Random hash sparse model.** We discussed the design of DSM and LSH models in Section 2. Here we present the details of the random hash model, where the sparsity pattern is determined by a random

hash of the input data (i.e. the same input data would always have the same sparsity pattern). The following code snippet shows the generation of a random mask that only depends on the input data using TensorFlow 2.x.

```
import tensorflow as tf

# seed: a fixed random seed
# inputs: the input tensor
# mask_dim: size of the masked tensor
# num_buckets: a large integer
# k: the dimension after masking

input_dim = inputs.shape[-1]
if input_dim != mask_dim:
  proj = tf.random.stateless_normal(
    shape=(input_dim, mask_dim),
    seed=seed)
  inputs = tf.einsum(
    '...i,io->...o', inputs, proj)
hs = tf.strings.to_hash_bucket_fast(
  tf.strings.as_string(inputs),
  num_buckets=num_buckets)
top_k_hash = tf.expand_dims(
  tf.nn.top_k(hs, k).values[..., -1],
  axis=-1)
mask = hs >= top_k_hash
```

### D.2 Learning random polynomials under other parameter settings

We present experiment results for learning random polynomial target functions with low intrinsic dimensions. To be precise, the target polynomial is $p(Ax)$, where $p$ is a polynomial of degree $d$ with sum of coefficient absolute value $< 1$, $x \in \mathbb{R}^n$ is a vector, $A \in \mathbb{R}^{k \times n}$ is a matrix with random orthonormal rows, and $n > k$. Note now the intrinsic dimension of the domain is $k$, while the inputs $x$ has higher dimension $n$. In Figure 7, we compare the mean squared loss for dense models and DSMs for $n = 64$, $k = 8$, and $d = 4$. We observe similar behavior as Figure 2, where the input dimension is the same as the intrinsic dimension. This validates our analysis in Section 3.

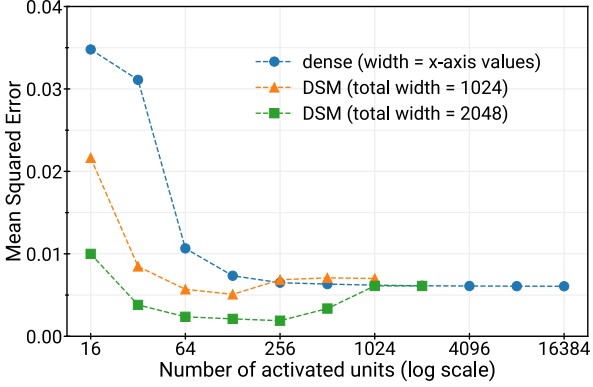

Figure 7: Scaling behavior of DSM compared with dense models for a random polynomial with low intrinsic dimensional domain. Similar to Figure 2, DSM outperforms dense models at the same number of activated units.