# OpenReview forum: "A Theoretical View on Sparsely Activated Networks"
_NeurIPS.cc/2022/Conference — NeurIPS 2022 Accept_

### Official Review · Reviewer_yqbh · 2022-06-30

**Rating:** 5
**Confidence:** 4
**Soundness:** 3 good
**Presentation:** 2 fair
**Contribution:** 2 fair

**Summary:**

This paper proposes the DSM model to sparsely approximate Lipschitz functions. The authors theoretically demonstrate their method in a wide range of scenarios, from one-layer shallow neural networks to switch and scale transformers. The original idea (but I am not sure as I am not familiar with this domain) of interpreting DSM as KNN is very interesting. However, :(, the experiment setting is a bit weak and seems to be finished in a rush.


####
I have increased the rating from 4 to 5 after rebuttal.

**Questions:**

Are there any potential competing methods in the related domain, especially those approaches like network pruning? Work like that may not worry about not getting sota results in performance, as it provides a new perspective for pruning networks. But providing a comparison with some competitors can make readers better understand the advantage and impact of this work.

**Limitations:**

1. I strongly encourage the authors to add figures to illustrate their concepts and ideas.
2. Experiments on SOTA neural networks will be much appreciated.
3. Needs a more detailed ablation study to justify the theory results.

**Strengths And Weaknesses:**

# Clarity:
## Strengths:
This paper offers a detailed introduction to the LSM model and other background knowledge.
## Weaknesses:
If the authors can further unify the usage of notations, the overall readability will be better. For example, the authors use s for the sparsity parameter, but in sec. 3.0 it switches to k, and later, k is used as the intrinsic dimension of input distributions.  The usage of notation A^x is also a bit confusing.

I also recommend the authors add some figures to illustrate their idea. For example, the Euclidean LSM and sec 3.0 can be well explained by figures.

# Originality:
I am not familiar with this domain, so I may not be able to judge this point.

But still, I find the argument in sec. 3.0 interesting. It points out a potential direction in that we may interpret neural networks as KNN operators. The current content can be enhanced in some directions. The authors may try to remove the constraints of unit B rows, typical network blocks like CNNs, Attention networks, and Residual connections, do not have such unit structures. Also, extending it to deep neural networks will be more attractive.

# Quality:
## Strengths:
The theory analysis is careful and in depth. But some settings and assumptions need either explanation to justify the necessity or adjustment to cater the practice demands.

## Weaknesses:
### Weird experiment setting:
1. Now that the main point is the efficiency of the proposed method, why not report inference time? FLOPS is a good metric but not enough.
2. Needs more detailed ablation study. Specifically, detailed study on how sparsity parameter s influence the model accuracy, approximation MSE, and inference time.
3. Now that the paper put much attention on discussing input distributions, the authors should also use input distributions on low dimensional manifold in R^n. Currently it is unclear how the input is sampled.
4. From numerical perspectives, polynomial may not be good choices, as they tend to be extreamly ill conditioned when degree is high. The authors may consider B-spline or Bézier curve for some realistic industrial scenarios. Also, random neural networks, even shallow ones, may be good candidates.

### Theoretical Settings need clarify and adjustments.
1. It is a bit weird to assume input distributions be uniform, can it be replaced as absultely continuous with respect to the uniform distribution (Lebesgue measure)?
2. Now that the proof is based on Euclidean LSH, it should be clearly stated in the theorems.

# Significance:
The theory results is good, but needs stronger empirical evidence to support it.

---

> ### Author Response · Authors · 2022-08-02
> **Response to Reviewer yqbh**
>
> Thank you for your review and the feedback provided. We first want to emphasize that our main contributions are our novel theoretical results and our principled exploration of the DSM model. Prior to our work, there was no formal model of sparsely-activated networks and no formal results on how to prove their ability to approximate functions. We urge you to reconsider your score based on this fact.
>
> 1. Clarity: We apologize for the notation confusion due to our usage of both ‘s’ and ‘k’ at different times to refer to the sparsity parameter. We will fix this and make the write-up more clear. We will also add figures to help illustrate our ideas of Euclidean LSH and the constructions in Section 3 in future versions of the paper. Given that the constructions are somewhat elementary, we hope that they are clear from the text, and we are happy to further clarify any questions you may have while trying to understand them.
> 2. Originality: We are glad that you found the connection of sparsely activated networks with kNN interesting. Although extending this connection to CNNs and deeper neural net architectures is an interesting direction, that would not fall under the scope of the current paper’s main message and hence we leave this exploration to future work. We are happy to discuss any concrete theoretical conjectures you have in this direction and see if they can be easily added to the paper.
> 3. Quality:
>     1. Reporting inference time in addition to FLOPs: While this is a good question, it is quite hard to develop a custom implementation that takes advantage of sparsity in our networks. We defer to the large-scale papers, such as Switch/Scaling Transformer, to see the inference time improvements of MoE methods.
>     2. “Needs a more detailed ablation study to justify the theory results.” We indeed performed such ablation studies in our paper. Please refer to Table 1 and Table 2 to see results of this sort. In an updated version of our paper, we add more results varying the sparsity/width.
>     3. Low-dimensional manifold input distributions in experiments: thank you for pointing this out! In Appendix D.2, we performed experiments with low-dimensional manifold input (where the inputs lie on a linear subspace of dimension 8 within a 64 dimensional input space) which demonstrate advantages of data-dependent sparsity. We plan to add more evaluations in this setting in an updated version.
>     4. Choice of Polynomials for the experiments: Note that ill-conditioned ness is not the problematic behavior for us as we aren’t trying to invert the polynomial to recover its coefficients. We merely want a good approximation for it. So as long as the polynomial is Lipschitz we are good. We ensure Lipschitzness by choosing the domain as $\[-1,1\]^d$ and restricting the polynomial coefficient vector to have $\|\|_1 \le 1$. Nonetheless, we have added experiments on random neural networks also as suggested and plan to perform experiments on B-spline based target functions in the future.
>     5. Generalizing from Uniform Distributions to Arbitrary Lebesgue measures: Note that since we focus on the $\|\|_{\infty}$ error in our theory, a non-uniform distribution would not make the problem easier either for the dense or sparse models as it is necessary to learn about the function locally in every part of the input space. For the $\|\|_2$ error, we believe this is an interesting direction and one we believe could be achieved in principle. This would complicate the theoretical argument however and we leave it for future work.
>     6. We will mention that the LSH we use is the Euclidean LSH clearly in our theorem statements.
> 4. Significance:
>     1. “Are there any potential competing methods in the related domain, especially those approaches like network pruning?”
>         Yes, in large-scale experimental papers, prior work already shows that MoE-based approaches are much more effective than network pruning at reducing the inference costs of large transformer models without sacrificing quality [Switch Transformer, Scaling Transformer, Vision MoE]. Our aim is not to provide more comparison to pruning, but instead, we theoretically explain why MoE models are effective. Our main contributions are theoretical. In this sense, we are not aware of other works which try to theoretically understand the power of MoE models.
>
> Additional Experiments: We define and evaluate a new target function: a randomly initialized 3-layer neural net. We scaled up the width up to 16384. (full set of results will be added to paper) We observe similar trends with this new target function that corroborate the experimental results in the original version of our paper.
> We have also performed new experiments, adding DSM/LSH layers to Vision Transformers evaluated on CIFAR-10 and observe similar trends.
> (Due to the limited time, we have not performed an extensive set of experiments in this space and plan to expand more in future versions of the paper.)

---

> > ### Comment · Reviewer_yqbh · 2022-08-03
> > **Thank you for the response.**
> >
> > Thank you for the response.
> >
> > I think there are still some problems to solve.
> >
> > **1. The main contribution of this paper is theoretical**
> >
> >  I agree that the majority contribution of this work is theoretical, but I am afraid certain experimental validations are still necessary. Please let me explain my concerns. The main reason is that some theoretical settings of this work do not match well with the practice. So they need either a detailed explanation or empirical validations to support why should we take them and whether taking them will not influence the generality of the proposed theorems.
> >
> > The theoretical settings that I think are too strong include:
> > 1. Lipschitz network assumption. This is a very strong assumption for realistic deep neural networks. DNNs are usually highly non-linear networks, and they are known to be suffered from unstable gradients.
> > 2. Uniform distribution assumption. This is too strong to stand in practice. We can hardly believe that real data can distribute uniformly. I think the paper may need to defend this choice more convincingly.
> > 3. Linear Subspace assumption. I think we can also hardly believe that data lie in some linear subspace of the Euclidean space. While the authors provide an extension to manifolds that are globally homeomorphic to linear subspace in the appendix, the results are not convincing as it omits the Lipschitz constant of the global homeomorphic function $f$ from the linear subspace to the k-manifold. Obviously, the property of this function should play an important role in the results.
> >
> > Besides, I am also curious about the statement "with probability>0.9" in the paper; which is the measure for this probability?
> >
> > Overall, I think it is OK to make any of these assumptions, but we may need further arguments to support that with these assumptions, the theoretical results can still be general enough for practical scenarios.
> >
> > **2. Reporting Inference times.**
> >
> > Can the authors explain why it is hard to take advantage of the sparsity in detail? I am a bit confused here.

---

> > > ### Author Response · Authors · 2022-08-06
> > > **Additional Clarifications.**
> > >
> > >
> > > We thank the reviewer for engaging with our response and for bringing up the additional questions. We address the concerns below.
> > >
> > > 1. Theoretical Assumptions:
> > >     1. Lipschitz network assumption: We want to point out that we **don’t** assume the network we use to learn the target function is Lipschitz. We only assume that the target function is Lipschitz. These are distinct assumptions. Coming to the assumption that the target function is Lipschitz, this is approximately satisfied in many reasonable settings in practice. For instance, the function which labels whether an image is a cat or not will not change its output when only a few pixel values change.
> > >     2. Uniform Distribution Assumption: While we agree this can potentially be a strong assumption, we want to emphasize that we only made this assumption to keep the theoretical calculations simple and convey the main message, i.e., the advantage in computational efficiency sparse models have over dense models. Importantly, this message continues to hold even when the uniformity assumption is abandoned. We present brief arguments why this happens here:
> > >         1. Under the $\|.\|_{\infty}$ metric we focus on in our theory, the regression procedure must estimate the function well in every local region of the space irrespective of the input distribution. This would imply that both the dense and sparse models would be required to see enough samples so that every sphere of radius $O(\epsilon)$ contains at least 1 sample. Both methods would still take the same number of samples and the sparse models would continue to retain their advantage in computational efficiency. Note that the main message is unaltered because we focus on computational efficiency savings and not on sample complexity savings in our work.
> > >         2. Even if we were to work with an $\|.\|_2$ metric as our target instead, we would need both models to approximate the target function well on a region of the input space which captures a large chunk of the probability and the samples required by both models would end up being similar up to constant factors. Hence the main message of our paper still holds.
> > >     3. Linear Subspace Assumption: As you point out we provide an extension to manifolds in the Appendix. We indeed have an assumption that the condition number of the Jacobian of the mapping from $\mathbb{R}^k$ to the $k$-dimensional manifold is constant and if this number is large then our bounds would blow up. However, it is very challenging and non-trivial to analyze general manifold surfaces without a bounded curvature as we assume. We wanted to shed light on why sparse networks are effective in the real world by abstracting away and simplifying some details. We believe some level of an abstraction is necessary for developing a first understanding. We aim to explore more complex geometries for future work.
> > >     4. "With probability >0.9": We apologize for the confusion caused by our writing style. This is a common way of writing statements which hold with high probability. In particular, for any $\delta>0$ we can have our statements hold with probability $> 1-\delta$ by paying a $\log(1/\delta)$ factor in the sample complexity and size of our models. We omit this $\delta$ variable for simplifying the result but will add a revised Theorem in the appendix which captures the dependence of the bounds on this parameter $\delta$.
> > > 2. Reporting Inference Times: The short reason is accelerators like GPUs are designed for dense matrix multiplications. Given a sparse matrix, they are unable to take advantage of the sparsity today unless the sparsity is “structured” somehow, e.g. block sparsity. Hence we wouldn’t see a speedup in wall clock times when we compare our LSH/DSM models (which use sparsity in an unstructured manner) to dense models of equivalent width. This issue of accelerators not being able to take advantage of unstructured sparsity also hinders the latency savings one obtains via the vast majority of weight pruning approaches such as Iterative Magnitude Pruning.
> > >
> > > We hope this clarifies and removes some of your concerns. Please let us know if you have any additional questions.

---

> > > > ### Comment · Reviewer_yqbh · 2022-08-08
> > > > **Thank you for the reponse.**
> > > >
> > > > Thank you for the response.
> > > >
> > > > **The uniform distribution assumption:**
> > > >
> > > > For a probability $\mu$, if $\mu(A)=0\Rightarrow U(A)=0,\forall A\in Borel(V)$, where $U$ is the uniform distribution on $V$, then it can satisfy your condition of at least one sample within each $O(\epsilon)$ ball. This generally corresponds to probability measures with positive densities on $V$. So I still cannot understand why the results cannot be extended to more general distributions.
> > > >
> > > > Further, if you require at least one sample within each $O(\epsilon)$ ball, then the total sample number will be proportional to $diam(V)^d$. (In the paper, the authors try to omit it by constraining the function input to $[-1,1]^d$, but we need to consider it when using the theory results to explain realistic problems.) This seems not well supported by the reality, where we only need a relatively small size of training data for various transformers and other deep networks.
> > > >
> > > >
> > > > **Linear subspace assumption:**
> > > >
> > > > The assumption of constant condition bijective function is too strong; this significantly lowers the influence of this theorem. Can it be changed into bounded condition numbers?

---

> > > > > ### Author Response · Authors · 2022-08-08
> > > > > **We can indeed extend our theory to more general distributions**
> > > > >
> > > > > Thanks for the response and additional questions.
> > > > > 1. Uniform distribution assumption: We want to clarify that our point we wanted to convey in our previous response was that indeed we **can** generalize our argument to more general distributions following a line of argument similar to what you propose. We chose not to do this for simplicity but will add this generalization in a revised version.
> > > > >
> > > > > 2. Exponential sample complexity: We understand that this does not capture reality exactly. For that reason we had the assumption of our data lying in a low-dimensional manifold within the $d$-dimensional space. This low-dimensional data manifold assumption is a widely believed property of real data and often the success of methods avoiding the curse of dimensionality is attributed to such an assumption. Moreover, our main point about the advantage of sparse networks over dense ones continues to hold both when the data manifold is high-dimensional or low-dimensional as the dense network requires an exponential number of samples as well. We hope this clarifies this issue and helps tie our theory more closely with what we see in practice.
> > > > >
> > > > > 3. Linear subspace assumption: Thank you for the question. If all we know is that the condition number is bounded, our approach will have to pay a potentially exponential in $d$ (the dimension of the ambient space) sample (and model size) complexity for both sparse and dense models and this will likely be unavoidable. We will try to work out the exact dependence of the bounds on the condition number in a revised version. This will help the reader understand how the performance degrades with the condition number.
> > > > >
> > > > > In addition, we want to present some evidence from other settings where analyzing linear spaces/linear functions has thrown light on more complex deep learning settings.
> > > > > 1. Understanding benign overfitting in overparameterized deep models by analyzing linear models: Benign overfitting in linear regression [Bartlett et al 2019], Surprises in High-Dimensional ridgeless least squares interpolation [Hastie et al 2019].
> > > > > 2. How transfer learning is able to transfer with few samples due to a shared (linear) geometry among the tasks: On the theory of Transfer Learning: The importance of task diversity [Tripuraneni et al 2020], Few-shot learning via learning the representation, provably [Du et al 2020], The benefit of multitask representation learning [Maurer et al 2016]
> > > > > 3. Understanding and mitigating multiple Descent phenomena by studying linear models: Optimal Regularization can Mitigate Double Descent [Nakkiran et al 2020], More data can hurt for linear regression: Sample-wise double descent [Nakkiran 2019].
> > > > > Hence, we see that analyzing linear spaces as a simplification can still yield useful and important insights and linearity often serves as a first model when trying to understand any phenomenon.
> > > > >
> > > > > We hope you can re-assess our paper in light of the above discussion. Thank you very much again for the thoughtful discussions and the timely engagement.

---

> > > > > > ### Comment · Reviewer_yqbh · 2022-08-09
> > > > > > **Thank you for the response.**
> > > > > >
> > > > > > Thank you for your patient response. The discussion period is about to end. In order to make the authors delightfully devote themselves to their other valuable work, I promise that I will respond to your urge to reconsider the rating. Please relax.
> > > > > >
> > > > > > But I may not be able to give the final rating yet, as there will be discussions between reviewers and the AC later.
> > > > > >
> > > > > > My final suggestions will be:
> > > > > > 1. Although you may lose something after considering different conditions, it will still be helpful to consider them. After all, a bijection with non-constant conditions is nothing but a concatenation of finitely many constant condition bijections.
> > > > > >
> > > > > > 2. This is the same for the distributions.
> > > > > >
> > > > > >
> > > > > > Please have a nice Tuesday or Wednesday, and take relax.

---

### Official Review · Reviewer_dumi · 2022-07-07

**Rating:** 6
**Confidence:** 3
**Soundness:** 3 good
**Presentation:** 3 good
**Contribution:** 3 good

**Summary:**

This paper provides a theoretical treatment of modern sparsely activated networks with the Data-dependent Sparse Model (DSM) model. The authors show that the DSM model can simulate modern sparsely activated networks and the locality sensitive hashing (LSH) model. It is proven in the paper that the LSH model can be expressive as a dense network for approximating real-valued Lipschitz functions while requiring much fewer FLOPs. Furthermore, experiments are conducted to validate the theoretical findings on Lipschitz target functions as well as the CIFAR-10 dataset.

**Questions:**

See weaknesses. I am not very sure how experiment results reflect the theoretical findings in the paper. I would appreciate it if the authors could explain it.

I am not very familiar with the field. Thus please correct me if I make any mistakes and I am happy to change my scores if my concerns/questions are addressed.

[Minor issues:]
1. Line 141: Why non-linearity \sigma is left out?
2. Line 143: I am not very sure why mask(x) is left out.
3. Line 272: maybe a typo 'qualtify'?

**Limitations:**

Yes, the authors have addressed the limitations and potential negative societal impact of the work.

**Strengths And Weaknesses:**

Strength:
1. The paper is the first work to treat sparsely activated networks theoretically, thus novel to me.
2. The paper is well-organized.

Weaknesses:
1. The theoretical analysis is based on the assumption of the L-Lipschitz target function and I am not sure how significant the work is. Furthermore, the neural network size used in the experiment is also very small.
2. I am not very sure about the relation between the theoretical findings and experiments. Theorem 4.1 and Theorem 4.3 conclude that LSH-based sparsely activated networks can be expressive as their dense counterparts when their size and number of samples match. As for size, the LSH model is measured using hash table size and the dense model is measured using width. As a result, in experiments, I am expecting to see the LSH model is as good as dense ones when # buckets == width of the dense network. However, in the figures, the width of dense networks is compared to the number of activated units.
3. For comparison of DSM and dense networks, the authors mention that 'Sparsity helps in both DSM and LSH models, ... using the same number of activated units.' However, it seems that the comparison may be unfair. Specifically, with 64 activated units, DSM chooses the best 64 units out of total of 1024 units while the dense one has only 64 units in total. It seems unsurprising to me that DSM is better than its dense counterpart.

---

> ### Author Response · Authors · 2022-08-02
> **Response to Reviewer dumi**
>
> Thank you for your review and feedback provided. We will fix all typos you pointed out and make a pass over the entire paper to fix any others remaining as well.
>
> 1. Lipschitz Condition: We believe the Lipschitz condition on the target ground truth is not always a very strong restriction. For instance, for image classification, the probability that an image is a cat changes slowly as pixel values change (rather than a sudden non-Lipschitz jump). We believe this assumption may not be strictly necessary. If we relax this assumption, it makes more sense to study DSM models where the bottom layer (routing layer) isn’t frozen after random initialization. This will allow the bottom layer to learn a possibly non-Lipschitz division of the input space into buckets which the top layer can then be used to fit on. However, theoretically analyzing the training dynamics in this non-convex setting are known to be very challenging and out of the scope of this paper.
> 2. Small neural net size: We would like to highlight here that the efficacy of MoE style approaches in deep transformers on large datasets has already been strongly established by numerous prior works. In a complex setting like that of a transformer, it is unclear where the advantage of MoE comes from due to many things going on at once. Hence, it is important to break down the model to a minimum working example where we can isolate and identify precisely what the advantage of a change like MoE is. That is what we did in our work and hence we focus on small and simple neural nets to analyze the effect of MoE. We understand that in real transformers, there might be other reasons why the MoE idea is beneficial. However, fundamentally the insights we exposited in our work still do apply. Exploring what leads to the benefits of MoE in deep transformer networks is perhaps out of the scope of theoretical analysis and in future work we aim to explore this question empirically.
> 3. Differing Pareto-Optimal Views: We can show the advantage of sparse models over dense models in two ways:
>     1. Show that sparse models achieving the same quality as a dense model are more efficient
>     2. Show that at the same amount of compute, a sparse model achieves better quality than a dense model.
>     We apologize for the confusion caused due to us adopting view (a) for our theory and view (b) for our experiments. For a stronger experiment-theory correspondence, we would like to highlight Figure 3, in which DSM and LSH models with fewer number of buckets than dense nets’ total width perform on-par or out-perform the dense nets’ quality, and the quality of DSM and LSH models only weakly depend on the number of activated units. This can also be seen from the following table, which compares the mean squared error for dense, DSM and LSH models (MSE for DSM and LSH models depends on number of activated units, so we sweep over number of activated units and reported the best and worst MSE). In Figure 2, quality has a stronger dependency on number of activated units, and shows a weaker correspondence with the theory (a potential reason might be the hidden constants in Theorem 4.1 and 4.3), for which we choose to compare the number of activated units since it reflects the computation cost better.
>
> | model / total width | 1024 | 2048 | 4096 | 8192 |
> | --- | --- | --- | --- | --- |
> | dense MSE | 0.0111 | 0.0108 | 0.0105 | 0.0102 |
> | DSM best MSE | 0.0081 | 0.0045 | 0.0027 | 0.0018 |
> | DSM worst MSE | 0.0117 | 0.0109 | 0.0098 | 0.0085 |
> | LSH best MSE | 0.0082 | 0.0053 | 0.0047 | 0.0053 |
> | LSH worst MSE | 0.0131 | 0.01045 | 0.0084 | 0.0081 |
>
> Additional Experiments: We have also performed the following additional experiments with the LSH/DSM models. We define and evaluate a new target function: a randomly initialized 3-layer neural net. We scaled up the width of the DSM and dense models up to 16384 (full set of results will be added to paper) We observe similar trends with this new target function that corroborate the experimental results in the original version of our paper.
> We have also performed new experiments, adding DSM/LSH layers to Vision Transformers. We evaluate on CIFAR-10, and we observe gains from the DSM and LSH methods over the dense model.
> (Due to the limited time, we have not performed an extensive set of experiments in this space and plan to expand more in future versions of the paper.)

---

> > ### Comment · Reviewer_dumi · 2022-08-05
> > **Response to the authors**
> >
> > I thank the authors for the clarifications. However, I am still concerned about the weak correspondence between the theory and the experiments, especially in Figure 2 as the authors mention. To me, the experiment results in Figure 2/3 do not actually convey much information as DSM and LSH models are obviously more complex than their dense counterparts.
> >
> > As for the table given,
> >
> > (1) Just want to make sure we are on the same page. The total number of buckets of DSM/LSH models is the same as dense models, is that right?
> >
> > (2) I see very little performance difference between dense and DSM/LSH models. The worst-case results of DSM/LSH are worse or almost indistinguishable from their dense counterparts.
> >
> > (3) The authors should mention what is the range of # the number of activations they sweep. And I suggest the authors report the number of active buckets for the best results, which may help other readers and me better interpret the results.

---

> > > ### Author Response · Authors · 2022-08-06
> > > **Experimental Clarifications**
> > >
> > > Thank you for engaging with our response and for the additional questions. We aim to address your questions and concerns raised below.
> > > For any fixed function that we are trying to learn, once the width of a dense model increases beyond a point we indeed see very limited amount of benefits in performance by going to sparse DSM/LSH models. This is to be expected however as the functions we fit in our experiments are simple functions which are easy to fit with a wide enough dense model. However, the behavior of the relative performance at lower widths (the left side of our graphs in Figures 2/3) is what we present as evidence of the efficacy of sparse models. Moreover, in practice on NLP datasets we commonly see today, the function to be learnt is sufficiently complex that even at very large widths we see a benefit in having a sparse model.
> > > As we increase the widths of the experts, the recent scaling law study on MoE models [Unified Scaling Laws for Routed Language Models. Clark et. al.] also observes that the benefit of having experts starts to diminish. So this behavior also happens with real language data at really large scales.
> > >
> > > 1. The total number of buckets of DSM/LSH models is the same as in dense models, correct.
> > > 2. As explained above, the right setting to focus on in Figures 2,3 is the space when the width of the models is not too large (which is the left end of the graphs) where we do see sizeable benefit to having sparsity.
> > > 3. We sweep a range of number of activated units all the way from 16 through 16384 (as shown on the x-axis of figures 2/3). Is this what you mean by active buckets?

---

> > > > ### Comment · Reviewer_dumi · 2022-08-07
> > > > **Response to the authors**
> > > >
> > > > I thank the authors for the clarification. Now I have a better understanding of the correspondence between the theory and the experiments.

---

> > > > > ### Comment · Reviewer_dumi · 2022-08-08
> > > > > **Score change**
> > > > >
> > > > > I have also increased my score from 5 to 6 to reflect the change.

---

### Official Review · Reviewer_R3gt · 2022-07-11

**Rating:** 6
**Confidence:** 3
**Soundness:** 3 good
**Presentation:** 3 good
**Contribution:** 3 good

**Summary:**

From my understanding, the main contribution of the paper is as follows:

1. the authors capture the sparsity structure of these popular transformers. They model the transformers into DSM models.
2. They show that the DSM model can represent the LSH model.
3. They provide theory on the LSH model. These theories can be used to interpret the success of Switch and Scaling Transformers.
4. Motivated by the theory, they proposed a new LSH-based model and run toy experiments to show its efficacy.





**Questions:**

See my comments below.

**Limitations:**

I have the following concerns:

1.  In the current  manuscript, the connection between contribution 1==>2==> 3==>4 is still a bit vague (see my comments in the **Summary** part) .
When reading the current version, it is easy to get confused about the main contribution of the paper: is it "explaining why general sparse model like Scale transformer works well."? Or is it " designing new LHS methods to save inference costs?"

For me, the contribution of the "explaining..." part outweighs the "designing.. " part. This is because the author didn't provide any real-data experiments on LSH. If no real-data experiment is provided, LSH is a pure theoretical tool to prove the theory on DSM and the "designing.. " part is minor. However, the current script over-emphasizes the "designing.." part, causing great confusion for me.

2. The performance of DSM on CIFAR-10 does not quite match the theory. To support the theory, I would suggest the authors run experiments on transformer-based NLP tasks instead of CV tasks on CIFAR.

3. All the theories are built on the function Lipschitz assumption. It would be better if the authors verify the Lipschitz condition. Is it a necessary condition or it is due to the limitation of the theory?  If it is the latter case, what is the main technical challenge to relax this assumption?


4. In line 255, why is random d-degree polynomial a Lipschitz function?


**Strengths And Weaknesses:**

See my comments below.

---

> ### Author Response · Authors · 2022-08-02
> **Response to Reviewer R3gt**
>
> Thanks for your review of our paper and your feedback. We apologize for the confusion caused by our writing between the two messages we have in our paper (1) explaining why general sparse methods like Switch transformer work well and (2) introducing new sparse methods like the LSH to save on inference costs. This is a very good point, that we will address in the next version.
>
> Our main motivation and contribution is (1). The LSH model is a particular instantiation of general sparse MoE models which we used to prove our theoretical results. We perform experiments using the LSH model to confirm our theoretical intuition. We do not intend for this to be the main contribution of our paper. We will update the paper to clarify our main message.
>
> Regarding the comment “theories are built on the function Lipschitz assumption.”
> Many real-world target functions exhibit (approximate) Lipschitz behavior, and hence, the condition is reasonable in many cases. For image classification, the probability that an image is a cat changes slowly as pixel values change (rather than a sudden non-Lipschitz jump).
>
> On the theoretical size, the Lipschitz assumption may not be strictly necessary. If we relax this assumption, it makes more sense to study DSM models where the bottom layer (routing layer) can be trained after random initialization. The bottom layer would learn a possibly non-Lipschitz division of the input space into buckets. Then, the top layer can fit a Lipschitz function on top of this. However, theoretically analyzing the training dynamics in this non-convex setting is known to be very challenging. We would need to prove that there exists an efficiently learnable representation, and that the target function can be Lipschitz on top of this. We believe this is a very interesting direction for future work.
>
> In line 255, we choose a random degree-d polynomial evaluated only on the bounded interval $\[-1,1\]^d$ and restrict the coefficient vector to have $\|\|_1$ smaller than 1, which makes the resulting function Lipschitz in the region of interest.
>
> Experiments on Transformer Based NLP Tasks: We thank the reviewer for the comment “I would suggest the authors run experiments on transformer-based NLP tasks instead of CV tasks on CIFAR.” However, due to the limited time, we were not able to add experiments on NLP tasks but we have performed experiments with adding DSM/LSH layers to Vision Transformers for CIFAR-10 classification and observe improvements with using sparsity (at the same number of FLOPs as a dense model). We will try to add experiments with Transformers on NLP tasks in future. We would also like to reiterate here that papers such as Switch/Scaling transformers are instantiations of data-dependent sparse models that show significant wins on NLP data.
>
>
> We have also performed new experiments, adding DSM/LSH layers to Vision Transformers. We evaluate on CIFAR-10, and we observe gains from the DSM and LSH methods over the dense model.
>
> | model | hidden dimension | # activated units | sparsity | test acc @1|
> | --- | --- | --- | --- | --- |
> | dense | 768 | 768 | 0% | 79.8 |
> | DSM-512 | 1024 | 512 | 50% | 79.9 |
> | DSM-384 | 1152 | 384 | 66.7% | 79.2 |
> | DSM-128 | 1408 | 128 | 90.9% | 80.1 |
> Table: Comparison of test accuracy on CIFAR-10 dataset for dense and DSM models with the same flops. We see sparse models with high sparsity and the same flops output-performs dense model.
>
> (Due to the limited time, we have not performed an extensive set of experiments in this space and plan to expand more in future versions of the paper.)

---

> ### Comment · Reviewer_R3gt · 2022-08-03
> **Could authors provide an updated version of the script?**
>
> I would like to thank authors for the detailed response. Also thanks for conducting additional experiments on ViT.  I notice that other reviewers have similar concerns about experimental results and theoretical settings (mostly about the Lipschitz condition).  Meanwhile, I still find it is an interesting idea to introduce "1==>2===>3 (see my **summary**)". It provides a good perspective on understanding modern large models.  I think this part of the contribution cannot be overlooked. I will keep my attitude of acceptance.
>
> Most of my concerns are addressed except for Concern 1. Could authors provide an updated version of the script?  I would like to check the revised version based on your response.  As a reviewer, it is my duty to make sure that the script can minimize the misunderstanding for general readers.
>
> Please use a different color to highlight the changes.

---

> > ### Author Response · Authors · 2022-08-05
> > **Revision Uploaded**
> >
> > Thank you for your response. We have uploaded a revised version of our manuscript with the changes highlighted in pink. Please let us know if this adequately addresses your concerns. We are happy to further incorporate any other feedback you have as well.

---

### Official Review · Reviewer_deSo · 2022-07-18

**Rating:** 6
**Confidence:** 3
**Soundness:** 2 fair
**Presentation:** 2 fair
**Contribution:** 2 fair

**Summary:**

The paper studies the Mixture of experts (MoE) architecture which has become popular in NLP recently as a way to increase the capacity of network without increasing depth.
The authors aim to develop a theoretical understanding of the MoE model/conditional computation. The authors begin with a formal model for conditionally activated sparse models which can capture common existing MoE models. The authors use LSH (locally sensitive hashing) for the gating in MoE and use this to derive a few theoretical results regarding the ability to approximate real valued Lipschitz functions in Rd.

The authors perform some small scale experiments to back up and verify their theoretical findings.

######## POST REBUTTAL ######
Thanks to the authors for running the experiments and for sharing the insights. Like I said earlier, it is important to study the theoretical underlining of MOEs. This paper starts with it, although as a researchers actively working in MOEs, I do not think that the paper exactly answers the key questions. The results proven are expected and not surprising, but on the other hand, as pointed by the authors, non-trivial to prove. So I would say that it is a decent paper at the moment and would suggest the authors to keep going in this direction to develop a more thorough understanding so that they can uncover some more fundamental results.

**Questions:**

I feel that the paper is quite borderline at the moment. It is important to have a theoretical understanding of the MoE models. The theorems prove some results which are expected and nothing surprising. On the other hand, they have not been proved. My suggestion would be to get more experiments to make the paper stronger.

**Limitations:**

Very well discussed.

**Strengths And Weaknesses:**

Strengths:
1. Relevant Problem
- MoEs are becoming very popular in NLP. Thus it is important to study their underlying theoretical and working mechanisms. The paper tackles this relevant problem.

2. LSH (locally sensitive hashing)
- Authors propose to use LSH for gating. This can actually be quite promising in my opinion because it takes the local vicinity into consideration.

3. Well written
- It is a very well written paper and is easy to follow
- I really like the limitations section. It is good to see for a change that there is somebody who knows and writes the limitations of their work.

Weaknesses:
1. Weak Experimental Evaluation
- I think that LSH is infact good. Authors need to perform more experiments to show its effectiveness.
- I am not suggesting to go to the huge model sizes but atleast medium scale models and datasets should be evaluated.

---

> ### Author Response · Authors · 2022-08-02
> **Response to Reviewer deSo**
>
> Thanks for your review of our paper and the feedback provided. While we agree that the LSH model has potential to be widely applicable in practice, our focus on this paper was on understanding the theoretical mechanisms behind the success of the MoE paradigm and formally proving that sparse models can match the approximation power of dense networks in general settings. LSH model is one particular instantiation of the MoE paradigm which we chose to be able to theoretically prove our results. While it may be possible to expand our work and to use LSH for SOTA large-scale networks, we believe that this is beyond the scope of our paper. Specifically, there are many components necessary to successfully train an MoE model, and isolating the improvements from an LSH-based routing function would be a very involved empirical undertaking. On the other hand, we believe that our empirical insights already generalize to more settings, as we discuss next.
>
> Additional Experiments: To address your comment “get more experiments to make the paper stronger,” we have performed additional experiments with the LSH/DSM models. We define and evaluate a new target function: a randomly initialized 3-layer neural net. We scaled up the width of the DSM and dense models up to 16384 (results up to width 4096 included here. full set of results will be added to paper). We observe similar trends with this new target function that corroborate the experimental results in the original version of our paper.
> MSE for regression on a random 3 layer NN target function with a dense model
> | model \ Width | 1024 | 2048 | 4096 |
> | --- | --- | --- | --- |
> | Dense MSE| 0.0000432 | 0.0000176 | 0.0000074 |
>
> MSE for regression on the same random 3 layer NN target function with sparse models of different sparsity levels and total widths
> |model \ Total Width| 1024 | 2048 | 4096 |
> | --- | --- | --- | --- |
> | DSM with sparsity 32 MSE | 0.00013 | 0.000070 | 0.000023 |
> | DSM with sparsity 64 MSE | 0.00011 | 0.000031 | 0.000019 |
> | DSM with sparsity 128 MSE | 0.00010 | 0.000042 | 0.000015 |
> | DSM with sparsity 256 MSE | 0.000087 | 0.000037 | 0.000017 |
> | DSM with sparsity 512 MSE | 0.000058 | 0.000038 | 0.000019 |
> | DSM with sparsity 1024 MSE | 0.000060 | 0.000021 | 0.000022 |
>
>
> We have also performed new experiments, adding DSM/LSH layers to Vision Transformers. We evaluate on CIFAR-10, and we observe gains from the DSM and LSH methods over the dense model.
>
> | model | hidden dimension | # activated units | sparsity | test acc @1|
> | --- | --- | --- | --- | --- |
> | dense | 768 | 768 | 0% | 79.8 |
> | DSM-512 | 1024 | 512 | 50% | 79.9 |
> | DSM-384 | 1152 | 384 | 66.7% | 79.2 |
> | DSM-128 | 1408 | 128 | 90.9% | 80.1 |
> Table: Comparison of test accuracy on CIFAR-10 dataset for dense and DSM models with the same flops. We see sparse models with high sparsity and the same flops output-performs dense model.
>
> (Due to the limited time, we have not performed an extensive set of experiments in this space and plan to expand more in future versions of the paper.)

---

> > ### Author Response · Authors · 2022-08-09
> > **Do the new experiments address your concerns?**
> >
> > From what we understand, your main concern/suggestion was "My suggestion would be to get more experiments to make the paper stronger."
> >
> > In the previous comment, we have added two new experiments:
> >
> > (1) the DSM model improves upon a dense model for larger model widths and for a third class of target function (3-layer networks)
> >
> > (2) adding sparse layers to vision transformers can retain the same accuracy as dense models having layers that are 90.9% sparse.
> >
> > Please let us know if these new experiments suffice to elevate your assessment above borderline.
> >
> > We also want to comment on your claim that "The theorems prove some results which are expected and nothing surprising."
> > We agree that the results are consistent with common wisdom, but we want to point out there is value in defining the model, identifying the precise theorem statements, and proving them.
> >
> > Moreover, the technical arguments are not immediately obvious. For example,
> >
> > (A) Lemma 4.2, which is the main technical tool in analyzing the LSH router, requires us to show that when we use K = O(k) hyperplanes, then we have that two points in the same LSH bucket are close in L2 distance with good probability. This analysis uses properties of the smallest singular value of random Gaussian matrices.
> >
> > (B) One of the observations behind Theorem 4.1 is that it is enough to simply output a constant value for each bucket. This is surprising, at least a priori, given that we are approximating a high-dimensional function, where the only restriction is that it is Lipschitz.
> >
> > (C) Our lower bounds (Theorems 4.3 and C.1) are also technically non-trivial and qualitatively informative.

---

### Meta-Review · Area_Chair_MSAu · 2022-08-30

**Recommendation:** Accept
**Confidence:** Less certain

**Metareview:**

The paper provides a theoretical analysis of sparsely activated neural networks. They introduce LSH (local sensitive hashing) as a new routing function for theoretical analysis and proved a few results on representation power and inference time. One reviewer pointed out that the theoretical results are expected and do not provide much interesting insight, which I agree with. Nevertheless, this is one of the early papers that study sparsely activated networks and may serve as a starting point. I recommend acceptance.


**Award:**

No

---

### Decision · Program_Chairs · 2022-09-14

Accept